

# Comparing high-latitude thermospheric winds from FPI and CHAMP accelerometer measurements

Anasuya Aruliah[1,] Matthias Förster[2], Rosie Hood[1], Ian McWhirter[1], Eelco Doornbos[3,4]

[1]Atmospheric Physics Laboratory, University College London, Gower Street, London, WC1E 6BT, UK
[2]Helmholtz-Zentrum Potsdam, GFZ German Research Centre for Geosciences, Telegrafenberg, 14473 Potsdam, Germany
[3]Old affiliation: Faculty of Aerospace Engineering, Delft University of Technology (TU Delft), Kluyverweg 1, 2629 HS Delft, The Netherlands
[4]Current affiliation: Royal Netherlands Meteorological Institute (KNMI), Utrechtseweg 297, 3731 GA De Bilt, The Netherlands

*Correspondence to*: Anasuya Aruliah (a.aruliah@ucl.ac.uk) and Matthias Förster (mfo@gfz-potsdam.de)

**Abstract**. It is generally assumed that horizontal wind velocities are independent of height above the $F_1$-region (> 300 km) due to the large viscosity of the upper thermosphere. This assumption is used to compare two completely different methods of thermospheric neutral wind observation, using two distinct locations in the high-latitude Northern Hemisphere. The measurements are from ground-based Fabry-Perot Interferometers (FPI), and from in-situ accelerometer measurements onboard the CHAMP satellite, which was in a near polar orbit. The UCL KEOPS FPI is located in the vicinity of the auroral oval at the ESRANGE site near Kiruna, Sweden (67.8°N, 20.4°E). The UCL Longyearbyen FPI is a polar cap site, located at the Kjell Henriksen Observatory on Svalbard (78.1°N, 16.0°E). The comparison is done in a statistical sense, comparing a longer time series obtained during nighttime hours in the winter months (November to January); with overflights of the CHAMP satellite between 2001 and 2007 over the observational sites, within ±2° (±220 km horizontal range). The FPI is assumed to measure the line-of-sight winds at ~240 km height, i.e. the peak emission height of the atomic oxygen 630.0 nm emission. The cross-track winds are derived from state-of-the-art precision accelerometer measurements at altitudes between 450 km (in 2001) to 330 km (in 2007); i.e. 100-200 km above the FPI wind observations. We show that CHAMP winds at high latitudes are systematically 1.5-2 times larger than FPI winds. In addition to testing the consistency of the different measurement approaches, the study aims to clarify the effects of viscosity on the height dependence of thermospheric winds.

## 1 Introduction

Global circulation models (GCM) of the upper atmosphere (80-600 km altitude) appear in two forms: climatologies based on empirical measurements, and theoretical models that calculate the atmospheric conditions using the principles of physics and chemistry. These models are important for space weather studies and are also applied in understanding and predicting drag on low-altitude satellites, space debris, and the study of re-entry of near Earth objects. The theoretical and empirical models rely on observations from ground-based instruments around the world and global observations by satellites to provide constraints and boundary conditions. In particular, models must account for energy from sources external to the upper atmosphere (i.e., direct solar radiation, particle precipitation and heat flow from above; radiative, conductive and convective



heating from below; the magnetospheric electrodynamic driver at high-latitudes), which is divided between acceleration of the gas and heating. The empirical evidence for the energy budget is provided by observations of winds and temperatures.

The use of accelerometers on satellites to measure thermospheric winds had previously been reported quite rarely (Marcos and Forbes, 1985; Forbes et al., 1993). Over the last few years CHAMP and GOCE winds have been reported (e.g. Förster et al., 2008; Doornbos et al., 2010). The advantage of this technique consists in the fairly direct in-situ measurement, with relatively high spatial (temporal) resolution, of the cross-track wind component along the orbital track with only a limited number of special assumptions for the data interpretation. Adding more satellites (e.g., GRACE and GRACE-FO), should allow better full wind vector reconstructions in terms of statistical averages (Förster et al., 2008; Förster et al., 2017) as well as parameterized statistical studies of the upper thermosphere dynamics in the near future. As a result, it makes it imperative that the derived winds are correct because satellites provide global coverage of the upper atmosphere, unlike the small number of ground-based instruments currently in existence. The larger databases and global coverage of the satellites will particularly influence a semi-empirical model such as the Horizontal Wind Model (Drob et al., 2015), which is commonly used as a climatology of winds to provide initial boundary conditions and validation for physics-based global circulation models (GCMs).

In this paper we show that upper thermospheric winds measured by the CHAMP satellites are systematically 1.5 to 2 times larger than those measured by ground-based Fabry-Perot Interferometers (FPIs) at an auroral site and polar cap site. It is imperative to know whether this discrepancy is real (i.e. there is a variation of speed with respect to height), or whether we have uncovered a problem of the absolute scaling of wind measurements by comparing FPIs with CHAMP. With incorrect scaling, there arises a problem of distortion of energy budget calculations of the upper atmosphere as demonstrated below.

Consider a simple argument where the added energy per unit volume is $\delta E$, and the wind measured by the satellite and FPI are $U_{sat}$, and $U_{FPI}$, respectively. The energy is redistributed between a change in kinetic energy and heating of the atmosphere. If the gas has density $\rho$, and is accelerated from being initially stationary to speed $U$, then the energy redistribution is given by Eq. (1)

$$\delta E = \frac{1}{2}\rho U^2 + \rho C_p \delta T \tag{1}$$

Let us assume that the satellite and FPI are measuring the same volume of gas, but are not absolutely calibrated. Let $\delta T_{sat}$ and $\delta T_{FPI}$ be the change in temperature due to heating as measured by the satellite and FPI, respectively. Thus the change in energy is given by Eq. (2), assuming a mean density $\rho$ during this process.

$$\delta E = \frac{1}{2}\rho U_{sat}^2 + \rho C_p \delta T_{sat} = \frac{1}{2}\rho U_{FPI}^2 + \rho C_p \delta T_{FPI} \tag{2}$$





Consider if the measurements from the satellite and FPI are such that $U_{sat} = 2U_{FPI}$, i.e. either the satellite or
the FPI (or both) are wrongly calibrated, then substituting for $U_{sat}$ in Eq. (2), and rearranging both equations,
leads to Eq. (3).

$$4\delta T_{FPI} - \delta T_{sat} = \frac{3\delta E}{\rho C_p} \quad \text{(3)}$$

Thus Eq. (3) demonstrates that for positive $\delta E$ (i.e. heating), the inferred satellite temperatures are larger than
the FPI temperatures (e.g. if $\delta E \approx 0$, then $\delta T_{sat} \approx 4\delta T_{FPI}$). In other words, the satellite wind measurements
imply that more energy is put into heating the gas, and less into accelerating the gas, while the FPI
measurements would indicate the reverse. This would result in a mismatch between modelled and observed
temperature changes. The FPI can measure temperatures to test this, as will be done in a future study. The
temperature discrepancy would also have a knock-on effect on the calculation of density $\rho$ of the gas as
determined by the satellites, or by ground-based FPIs, since $\delta\rho = nk_B\delta T$, where $k_B$ is the Boltzmann constant and
$n$ is the number density of the gas particles. Note that this argument for a point measurement is oversimplified.
Owing to the high viscosity and heat conductivity, the whole air column above the measurement location should
be accelerated and heated.

**2 The CHAMP accelerometer data**

The challenging mini-satellite payload (CHAMP) was managed by the GFZ German Research Centre for
Geosciences of the Helmholtz Centre Potsdam. This mission was designed to perform detailed studies of the
Earth's gravitational and magnetic field with unprecedented accuracies and space/time resolutions as well as
GPS atmosphere and ionosphere profiling. The spacecraft was launched in July 2000 into a circular near-polar
orbit with 87.3° inclination at an initial altitude of ~460 km (Reigber et al., 2002). Its orbital altitude gradually
decayed to ~400 km in 2003 and ~330 km in 2008, and ended in September 2010.
One key scientific instrument onboard CHAMP was a triaxial accelerometer. It was located at the spacecraft's
center of mass and effectively probed the in-situ air drag. Thermospheric mass density and cross-track neutral
wind can be obtained from the drag acceleration observations. It is very difficult to determine the error estimate
because it depends on several variables as discussed in Doornbos et al. (2010) and shown in Table 5 from Visser
et al. (2019). This indicates that for force-derived winds, the largest sensitivity is to energy accommodation,
which is of the order of several tens of ms$^{-1}$.
A first analysis of the high-latitude thermospheric wind circulation in dependence on the IMF orientation was
performed by Förster et al. (2008) using the preliminary methodology of cross-track wind estimations from
accelerometer data as described in Liu et al. (2006). Förster et al. (2011) then presented an overview of the
average transpolar thermospheric circulation in terms of the vorticity. Here, they made use of the newly



calibrated and re-analysed data set that resulted from an ESA study, initiated for the Swarm satellites mission
launched in November 2013 (Helleputte et al., 2009).

As pointed out by Doornbos et al. (2010), the along-track wind is not resolvable because it induces a similar
signal in the acceleration as the density variation. This wind component is ignored, or the value from an
empirical wind model is used, because the along-track wind is a relatively small magnitude in comparison with
the satellite speed of 7.6 km s$^{-1}$. The empirical wind model used is, for example, HWM90 (Hedin et al., 1991) or
its latest edition HWM14, as published by Drob et al. (2015). In polar areas the along-track wind velocity can
achieve up to 10% of the satellite speed. Consequently, the along-track mass density estimation can have an
error of about 20% in the polar latitudes, because the acceleration is proportional to the wind velocity squared
(e.g., Doornbos et al., 2010). But it is less easy to estimate the error in the cross-track wind in the polar region
due to considerably smaller acceleration signals. There are also systematic contributions from other sources such
as gas-surface interactions, surface properties, spacecraft shape, spacecraft attitude and radiation pressure
accelerations, which make the satellite aerodynamic coefficients difficult to resolve (see Doornbos et al., 2010,
and the error budget in Appendix A of that paper; Mehta et al., 2017 and March et al., 2018). The pre-processed
data of the accelerometer were re-sampled to 10-sec averages for the further use in this study. Measurements of
10-sec cadence correspond to a spatial separation of 76 km or about 2/3° in latitude between the individual data
points.

**3 The Fabry-Perot Interferometer data**
The advantage of using Fabry-Perot Interferometers is that they make direct measurements of thermospheric
wind speeds using only a few instrumental or geophysical assumptions. They are also generally reliable
instruments that can be left to run for months at a time. The FPIs operated by University College London are
located at the Kiruna Esrange Optical Platform System (KEOPS) in northern Sweden; and on the island of
Svalbard at the Adventdalen Observatory (before November 2006), from which it was moved to the Kjell
Henriksen Observatory (after November 2006). The geographic and geomagnetic coordinates of these two
stations are given in Table 1 for Kiruna (KEOPS) and Table 2 for Longyearbyen (Svalbard). The Altitude-
Adjusted    Corrected    Geomagnetic    Coordinates    (AACGM)    are    obtained    from
http://sdnet.thayer.dartmouth.edu/aacgm/aacgm_calc.php#AACGM (Shepherd, 2014). A date of 15 December
2002 was used, for an altitude of 240 km. Owing to the large field-of-view of the FPIs (see later) the locations of
the volumes observed by the FPIs in the East and West look directions are also given, and the corresponding
MLT.









**Table 1**

| FPI Site | Geographic coordinates | AACGM geomagnetic coordinates for 15 Dec 2002 at 240 km altitude | AACGM magnetic local time (MLT) midnight in UT |
|---|---|---|---|
| Kiruna (KEOPS) | 67.87°N, 21.03°E | 65.08°N, 103.32°E | 1.860 hrs |
| Kiruna (KEOPS) EAST | 67.87°N, 26.6°E | 64.8°N, 107.8°E | 2.16 UT |
| Kiruna (KEOPS) WEST | 67.87°N, 15.5°E | 65.4°N, 98.9°E | 1.57 UT |


**Table 2**

| FPI Site | Geographic coordinates | AACGM geomagnetic coordinates for 15 Dec 2002 at 240 km altitude | AACGM magnetic local time (MLT) midnight in UT |
|---|---|---|---|
| Longyearbyen (KHO after 2006) | 78.15°N, 16.04°E | 75.38°N, 111.80°E | 2.43 UT |
| Longyearbyen EAST (KHO after 2006) | 78.15°N, 33.6°E | 74.4°N, 123.6°E | 3.21 UT |
| Longyearbyen WEST (KHO after 2006) | 78.15°N, -1.5°E | 76.8°N, 100.3°E | 1.66 UT |
| Longyearbyen (Advendalen before 2006) | 78.19°N, 15.92°E | 75.43°N, 111.80°E | 2.43 UT |
| Longyearbyen EAST (Advendalen before 2006) | 78.19°N, 33.5°E | 74.4°N, 123.6°E | 3.21 UT |
| Longyearbyen WEST (Advendalen before 2006) | 78.19°N, -1.7°E | 76.9°N, 100.3°E | 1.66 UT |


A significant limitation of ground based FPIs is that optical measurements of airglow and aurora at
thermospheric altitudes are only possible during the night when the sun's zenith angle is greater than 98°. This
means that the high latitude FPI observing season runs only in the winter months: from September to April at
KEOPS; and October to March at Longyearbyen. The FPIs have been nearly continually observing the 630 nm
emission from airglow and aurora every winter night since 1981 and 1986, respectively. Complete 24 hours of
observation are possible during November to January at Longyearbyen. Thermospheric winds have been
monitored by calculating the Doppler shifts of the 630nm airglow radiation intensities. The FPI instrument has a
mirror that rotates to look in several directions (e.g.north, north-east, east, south, west, north-west, zenith and a
calibration lamp) to provide line-of-sight wind measurements at a fixed elevation angle. The exposure times can
be as low as 10 seconds, up to 120 seconds. A typical complete scan cycle takes ~4 minutes for Kiruna and ~5



mins for Longyearbyen. After 1999, when laser calibrations were made possible, thermospheric temperatures
were measured from the thermal broadening of the emission line. More details of operation may be found in
Aruliah et al. (2005) and references therein.
The 630nm emission has a peak intensity at an altitude of around 240 km. So measurements of the Doppler
shifts and thermal broadening of the emission line are used to determine the winds and neutral temperatures of
the upper thermosphere (> 200km altitude). The elevation angle of the mirror is 45° for the Kiruna FPI and 30°
for the Longyearbyen FPI. Thus the radius of the field-of-view is 240 km and 416 km, respectively, which
represents roughly a 5° and 8° separation in latitude of the north and south viewing volumes at the respective
sites. At these high latitudes where the magnetospheric dynamo dominates the plasma flows, ion-neutral
coupling can create meso-scale structures in the upper thermosphere on horizontal scale sizes of as little as ~100
km (e.g. Aruliah et al., 2001, Emmert et al., 2006a). So average wind speeds have been determined for each of
the 4 cardinal look directions in order that the meso-scale structure is not lost. The winds are strongly dependent
on UT, season, solar cycle and geomagnetic activity due to the dominant forcing mechanisms of pressure
gradients and ion-neutral coupling in the high latitude upper thermosphere. The maximum average wind vector
magnitudes measured by an FPI at Kiruna were shown to be in the range 100-300 $ms^{-1}$ and the errors of
measurements were around 10-20 $ms^{-1}$ (Aruliah et al., 1996). The main sources of error are:

a)   Poor signal to noise when the 630 nm intensities are low, such as at solar minimum, or geomagnetically

quiet conditions.

b)   The existence of large vertical winds. These break the assumption that the winds are predominantly

horizontal. Vertical winds are generally small, but can be a few 10s of $ms^{-1}$ at high latitudes (Aruliah

and Rees, 1995; Ronksley, 2016). Large vertical winds introduce an error of a few per cent into the

calculation of a horizontal wind component from the line-of-sight measurement.

c)   The assumption that the neutral winds are nearly constant with respect to altitude above 200 km owing

to the very low density and consequently high viscosity of the upper thermosphere.

The altitude distribution of the 630 nm emission has a peak emission altitude of between 220-250 km (e.g. Link
and Cogger, 1988; Vlasov et al., 2005). However, the emission profile also has a full width at half maximum
intensity of around 50-70 km, i.e. sampling altitudes tens of km below and above the emission peak. The
ground-based FPI observes a height-integration of the emission along the line-of-sight. The measured Doppler
shift is therefore an integration of the Doppler shifts at all altitudes, weighted by the emission profile. However,
there are several reasons to justify why we are confident that the FPI provides a good sample of the winds at
~240 km altitude. The excited atomic oxygen state in the O ($^1D$-$^3P$) transition is a forbidden transition with a
long life-time of ~110 sec (Bauer, 1973), which allows the atom to thermalise before emission and be
representative of the surrounding gas. Below 200 km the molecular composition increases significantly, and the
long lifetime means that the 630 nm emission is quenched due to molecular collisions with $N_2$ and $O_2$.
Consequently we can assume there is minimal contribution of Doppler shifts from below 200 km altitude, which
is a region where the neutral wind magnitude has a large height dependence (note that the winds at 100 km
altitude are a few tens of $ms^{-1}$ while at 250 km altitude are a few hundreds of $ms^{-1}$). Above the altitude of the





emission peak the flux falls off rapidly with altitude and also with distance from the FPI, which minimises the
contribution of winds from the region above. Overall it is suggested that the FPI measured winds may
underestimate the winds at ~240 km altitude by no more than about 10% due to the contribution of winds below
the peak emission height.
**4 CMAT2 model winds**
The UCL Coupled Middle Atmosphere Thermosphere (CMAT2) model is a 3-dimensional, time-dependent
physics-based model, that solves numerically the non-linear coupled continuity equations of mass, momentum
and energy (Harris et al., 2002). The model has a latitude resolution of 2°, longitude 18°, and a one third scale
height for a height range of ~15 km (top of the troposphere) to 300-600km (top of the thermosphere).
Thermospheric heating, photodissociation and photoionisation are calculated for solar X-ray, EUV and UV
radiation between 0.1-194 nm (Fuller-Rowell, 1992; Torr et al, 1980a and 1980b; Roble, 1987). High latitude
ionospheric parameters of ion and electron densities and temperatures, plus field-aligned plasma velocities, are
from the Coupled Sheffield University High-latitude Ionosphere Model (Quegan et al., 1982; Fuller-Rowell et
al., 1996). The high latitude auroral precipitation is provided by the TIROS/NOAA auroral precipitation model
(Fuller-Rowell and Evans, 1987) and the high latitude electric field model is from Foster et al., (1986). Other
features are detailed in Harris et al. (2002). The CMAT2 winds will be presented as part of the discussion
below.

**5. Results**
Data were chosen from the 3-year periods 2001-2003 and 2005-2007, when the CHAMP satellite was in orbit.
These represent periods of solar maximum and minimum, respectively. CHAMP data were collected all year
around, but the FPI data were limited to nighttime periods only.
**5.1 CHAMP average winds**
Figures 1 and 2 show plots of average CHAMP accelerometer measurements of the cross-track thermospheric
wind component during the whole period of years 2001–2003, that were obtained during direct overflights
above the FPI stations at Longyearbyen (Figure 1) and KEOPS (Figure 2). The cross-track wind component is
defined as pointing into positive y-direction of the S/C coordinate system with its x axis along the orbital trace,
the z axis toward nadir, and the y-axis completing the right-hand system. The y-wind component therefore
points perpendicular to the orbital plane to the right side, when looking in the direction of flight. Given the high
inclination (87.3°) of the CHAMP satellite, this corresponds approximately to the geographically eastward
direction for the ascending orbital track (blue lines in Figure 1) except for very high geographic latitudes (see
below). The cross-track wind measurements of the descending orbital tracks (red lines) have been flipped in sign
to get nearly the same eastward wind component.

Figure 1a and 2a show average values for all data, while Figures 1b, 1c, 2b and 2c show the summer and winter
averages. There are many more data points for the Longyearbyen station, at a higher geographic latitude
compared with KEOPS. This confirms the fact that the relative probability of overhead crossings of high-



latitude stations by low Earth orbiting (LEO) satellites with a near-polar circular orbit augments with increasing latitude. A statistical study has to make some compromise with respect to the area of accepted local coincidences of the satellite recordings above the ground-based observations and also with regard to the further data binning. Here, a circular area with 2° radius and hourly bins versus local time have been used which produces a sufficiently good coverage. Further binnings have been been tested to investigate the effect on the results. A shorter radius deteriorates the statistics within the bin, while larger bins tend to smear the spatial and temporal variations. Data filtering with respect to other parameters like, e.g., season, solar wind and interplanetary magnetic field (IMF) values, solar radiation and geomagnetic activity indices should be taken into account if they appear to make a significant effect.

The variance of the cross-track neutral wind magnitude is considerably larger during the whole day above the station at higher latitudes. The average phase of the diurnal eastward wind variation differs also considerably between the data sets of the two observatories. The eastward wind maximizes during the pre-midnight hours over Longyearbyen, while a smaller maximum and a shorter interval of eastward wind is seen at lower latitudes above the KEOPS station (2–3 hours versus about 6 hours). The eastward neutral thermospheric wind is approximately sinusoidal for Longyearbyen (Fig. 1a), but reveals two maxima/minima over the KEOPS station (Fig. 2a). The westward wind maximizes there at about 19 LT and prior to midday (~11 LT). Finally, the variance of the cross-track neutral wind magnitude over the lower latitude station KEOPS is relatively large during the afternoon to early nighttime hours (~15–20 LT). This might be due to the position of this station relative to the large dusk cell which is known to be strongly dependent on, in particular, the IMF By component (cf., e.g., Rees et al., 1986; Killeen et al., 1995; Förster et al., 2008 and Förster et al., 2011). In contrast to KEOPS, the higher latitude station Longyearbyen is located close to or even poleward of the dusk cell's focus, so that the cross-polar cap circulation of the neutral thermospheric air dominates.



**Figure 1: CHAMP observations over Longyearbyen during solar maximum 2001-2003. a) All data – ascending (blue)**
**and descending (red) averages; b) Summer (May-Aug) and c) Winter (end Oct-early Mar).**

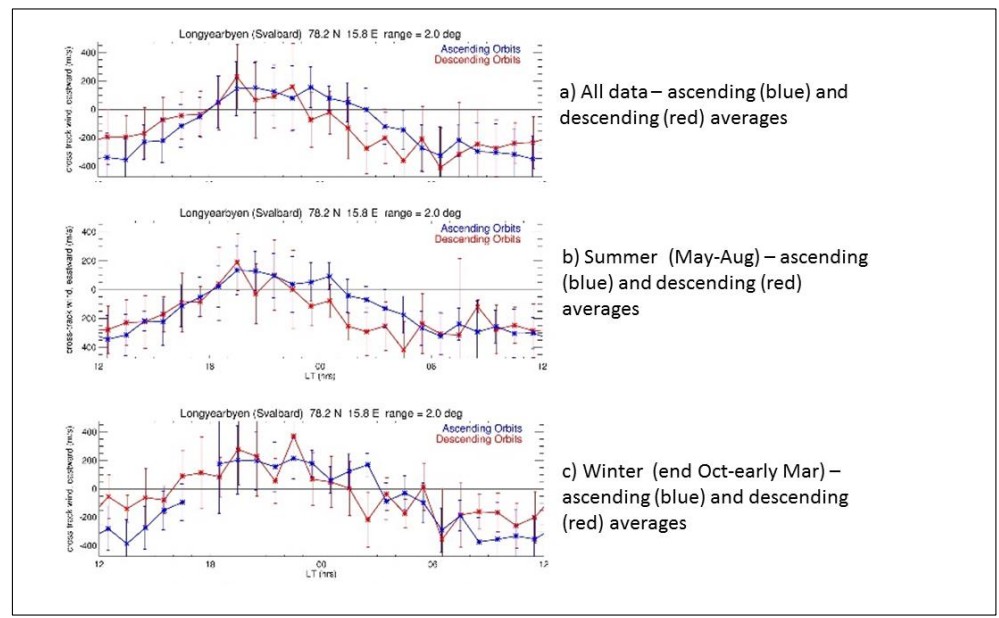


**Figure 2: CHAMP observations over KEOPS during solar maximum 2001-2003. a) All data – ascending (blue) and**
**descending (red) averages; b) Summer (May-Aug) and c) Winter (end Oct-early Mar).**

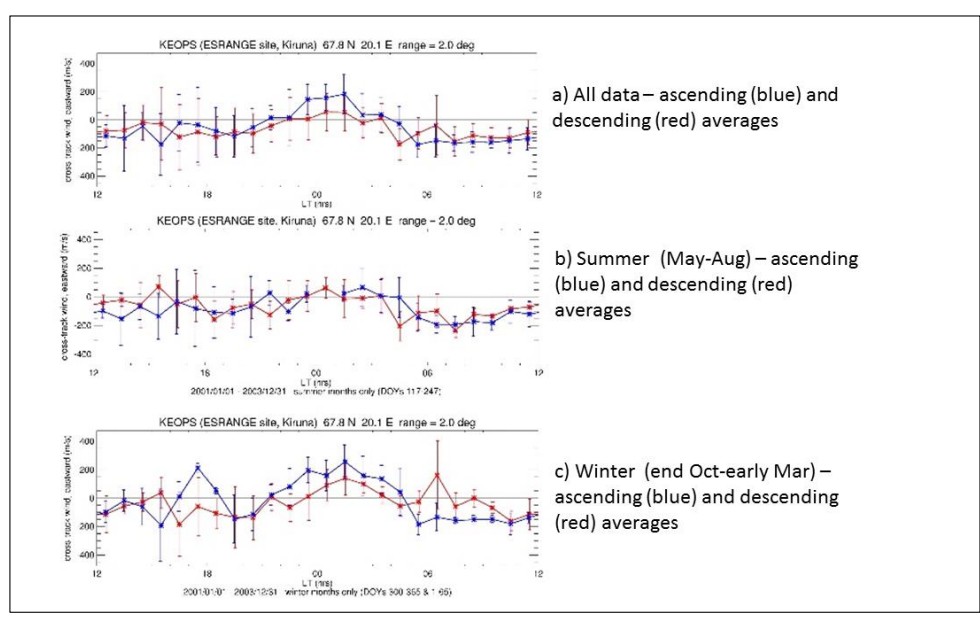



Figures 1b and 1c repeat the statistical plot of Fig. 1a for Longyearbyen for the years 2001-2003 with high solar
activity, but confined to the winter and summer months, respectively. Similarly for Figures 2b and 2c for
KEOPS. The statistical significance is therefore reduced, in particular for the KEOPS station, but seems to be
still sufficient. The winter recordings (Fig. 1c and 2c) during nighttime hours can directly be compared with the
FPI observations (Figs. 5 and 6).
The principle behaviour of the Longyearbyen and KEOPS eastward wind component is similar to that for the
full year coverage, but there are also obvious seasonal differences. The wind component amplitudes, in
particular the eastward maxima, are smaller during summer compared with the winter months, while the phases
are almost the same. The statements about the variance of the eastward/westward wind component for both
stations that have been made with respect to the full year statistics in Fig. 1 hold also for both winter and
summer plots, maybe with slightly larger values for the winter months.
The ascending and descending orbits are analyzed separately in their statistical behaviour (blue and red lines and
vertical bars, respectively), and show distinct differences. This points to the problem of co-alignment of the
ascending and descending orbital tracks (despite the simple sign flip). The small offset of ~2.7° from a strict
polar orbit of the satellite causes some deviation from the east/westward pointing of the cross-track
measurements. At low to mid-latitudes, the deviation from purely geographically eastward direction corresponds
in good approximation to this colatitude angle of the satellite's inclination $\beta \approx 2.7°$, but at high latitudes and in
particular near the poles it can deviate considerably. This non-alignment angle $\alpha$ (deviation from purely
eastward) can be estimated in dependence on the observer's colatitude $\theta$ with spherical angle relations using a
simplified spheric geometry of the Earth as in Eq. (4).
$$\alpha = \arcsin\left(\frac{\sin \beta}{\sin \theta}\right) \tag{4}$$

Using the geographic coordinates of the observatories in Table 1, one gets an $\alpha$-angle of 7.2° and 13.3° for
KEOPS and Longyearbyen, respectively. The angular difference ($2 \cdot \alpha$) between the two one-component cross-
track wind measurements of the ascending and descending orbital tracks is already considerable for the most
northward station at Longyearbyen and this offset can be noticed in, for example, Figure 1a as an offset between
the wind averages for ascending and descending orbital tracks during certain intervals, where the wind
component perpendicular to the zonal wind direction, i.e. the north-south meridional wind, is large. This is
obviously the case for the nighttime hours between 23–05 LT and the daytime hours between ~10–17 LT for
Longyearbyen and for a few nighttime hours between 23–03 LT for the KEOPS FPI station.
If the FPI technique, in particular the tri-static measurements for certain periods (Aruliah et al.,2005; Griffin et
al., 2008), allow the determination of specified neutral wind directions, one might consider comparing the wind
magnitudes for the descending and ascending orbital tracks separately for an eastward $\pm\alpha$ orientation,
respectively. Here, one should note, that at an observation point with an even higher geographic latitude (ideally
at ~86.2° geographic latitude, where the two branches of one-component observations would be perpendicular



to each other) it would in principle be possible to derive the full thermospheric wind vector from the cross-track
accelerometer measurements. This is, strictly spoken, valid in a statistical mean with characteristic times of a
few days, i.e. with the repetition period of ascending and descending orbits over one and the same high-latitude
location.

The meridional component is much larger than the zonal one during considerable periods of the nighttime
observation. So, to minimize the error in comparing the neutral wind magnitude, it would be better to compare
the full vectors. Already a small error of the measurement orientation could make a large effect on the relatively
small eastward wind component, which could lead us to wrong conclusions about the characteristics of the
differences between FPI and CHAMP accelerometer measurements. The offset between the geographic and
geomagnetic coordinates allows the construction of the full vector plots as statistical averages taken over a
period of at least 131 days of CHAMP's precession period in order to cover all local times. This statistical
mapping is limited to magnetic latitudes poleward of about > 60° for both hemispheres (cf. Förster et al., 2008).

**Figure 3: CHAMP observations over Longyearbyen during solar minimum 2005-2007. a) All data – ascending (blue)**
**and descending (red) averages; b) Summer (May-Aug) and c) Winter (end Oct-early Mar).**

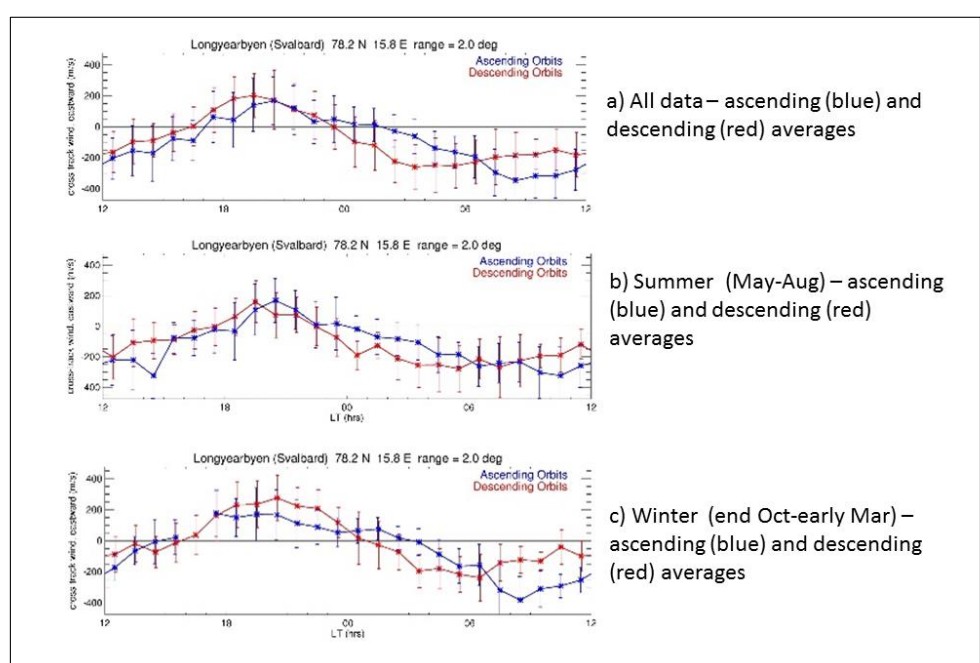






**Figure 4: CHAMP observations over KEOPS during solar minimum 2005-2007. a) All data – ascending (blue) and**
**descending (red) averages; b) Summer (May-Aug) and c) Winter (end Oct-early Mar).**

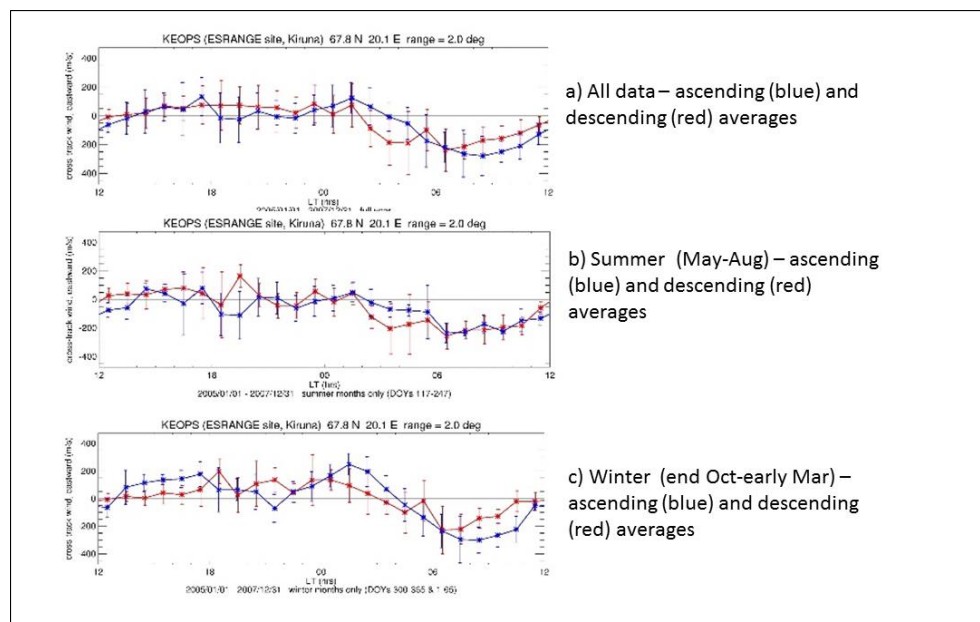


Figures 3 and 4 show the corresponding plots to Figs. 1 and 2, but now for low solar activity conditions during
the years 2005-2007. They reveal some differences as, e.g., generally smaller amplitudes and different wind
phases. Here, the zonal wind above KEOPS seems to point eastward during most times of the day except the
morning hours 04–11 LT.


**5.2 FPI average winds**
The average winds observed at Kiruna and Longyearbyen are presented in Figures 5 and 6. Local Time is 1 hour
ahead of Universal Time for both sites. The format of these figures is that the plots 5a, 5c, 6a and 6c show the
average zonal wind component, comparing observations to the East and West of the site. The plots 5b, 5d, 6b
and 6d show the average meridional wind component from the North and South look directions. The full set of
cardinal direction measurements, are presented to provide a context for the comparison with the zonal wind
measurements made by CHAMP, especially since the CHAMP y-axis is only roughly zonal. The standard error
of the mean ε is added as an error bar to the East and North data, where $\varepsilon = \sigma/\sqrt{(N-1)}$, $\sigma$ is the standard deviation
and $N$ is the number of data points. The periods of data cover the winter months of 2001-2003 and 2005-2007 to
match with the CHAMP datasets. The FPIs cannot measure winds during cloudy periods owing to the scatter of
light by the clouds, and are only able to observe the emission during the hours of darkness. Thus the observing
days cannot be identical to the dates when CHAMP passed overhead of the two sites. Longyearbyen has 24
hours of darkness during the months of November to January, so there are nearly 24 hours of observations, but



the longest period of darkness at Kiruna is around 18 hours in mid-winter. CHAMP, meanwhile, is able to
provide a full 24 hours of observations from drag measurements.
There are consistent differences in the winds observed to the geographic East and West, or to the North and
South. This is understandable because the Kiruna site is, on average, at the equatorward edge of the auroral oval,
while Longyearbyen is at the poleward edge. The expansion and contraction of the auroral oval during an active
period means that the northern half of the FPI field-of-view can be very different from the southern half. In fact,
Emmert et al. (2006b) have shown that high latitude neutral winds are better ordered in geomagnetic coordinates
of magnetic latitude and magnetic local time than in geographic coordinates and universal time. The AACGM
geomagnetic coordinates shown in Tables 1 and 2 give an indication of how different are the magnetic latitudes
for the East and West look directions.
Figure 5 shows average zonal and meridional winds from FPI observations at Longyearbyen. Figs 5a and 5b
show solar maximum years (2001-2003), while 5c and 5d show solar minimum years (2005-2007). Figs 5a and
5c show the zonal winds to the East and West using the convention of +East, while Figure 5b and 5d shows the
meridional winds to the North and South, using +North.






**Fig 5 FPI winter average wind components at Longyearbyen for geomagnetically quiet conditions (0<Kp<2). The**
**standard error of the mean is shown for one example component. Solar maximum (2001-2003) a) zonal, b) meridional**
**average winds. Solar minimum (2005-2007) c) zonal, d) meridional average winds. North and East are purple lines,**
**while South and West are light blue**.



Fig 5 FPI winter average wind components at Longyearbyen for geomagnetically quiet conditions (0<Kp<2). The standard error of the mean is shown for one example component.

Solar maximum (2001-2003)
a) Zonal average winds
b) Meridional average winds

Fig 5 FPI winter average wind components at Longyearbyen for geomagnetically quiet conditions (0<Kp<2). The standard error of the mean is shown for one example component.

Solar minimum (2005-2007)
c) Zonal average winds
d) Meridional average winds



Longyearbyen is just within the polar cap. The winds are predominantly antisunward despite the geomagnetic
activity level, since this is the direction for both the pressure gradient and ionospheric convection. As a result,
Longyearbyen observations are a somewhat less obvious indicator of ion-neutral coupling behaviour than
observations at KEOPS in the period 1800-2100 UT. The Longyearbyen solar maximum (2001-2003) winter
winds (day numbers 300-65), during geomagnetically quiet conditions ($0 \leq Kp < 2o$) are shown in Figures 5a
and 5b. The zonal winds (Figure 5a) show westward winds before 1800 UT and then eastward winds for ~6
hours which then turn westward. The maximum wind speed is about 200 ms$^{-1}$ eastward between 1800-2400 UT.
The meridional winds (Figure 5b) are slightly northward before 1700 UT, then turn southward until 0600 UT,
and return northward. The maximum speed is about 200 ms$^{-1}$ southward at about 0100 UT. The standard errors
of the mean are around ±30 ms$^{-1}$, however the values vary systematically through the night. Between 1800-2100
UT the standard error is around 3 times larger than between 0300-0900 UT when it is very small.
Figures 5c and 5d show the Longyearbyen FPI winds for clear nights during winter (DOY 300-65) 2005-2007,
geomagnetically quiet conditions ($0 \leq Kp < 2o$). There is a full 24 hours of observations in this dataset, and the
extreme quiet of this solar minimum period has provided a large number of observations for this category. The
antisunward flow appears clearly. There is a strong phase lag between the observations to the North and South.
This is puzzling because it cannot be explained in terms of ordering high latitude winds in geomagnetic
coordinates. This category is for the most geomagnetically quiet conditions possible: solar minimum during a
prolonged solar minimum, and the lowest Kp values. Under these conditions the geographic coordinate system
under which the solar flux heating operates, should be the most appropriate. The standard errors of the mean are
very small, around ±10 ms$^{-1}$, though again there is a systematic trend. Between 2100-0300 UT the standard error
of the meridional wind is about 2-3 times larger than at other times. Between 1500-2000 UT the zonal wind
standard error becomes considerably larger.



**Fig 6 FPI winter average wind components at KEOPS for geomagnetically quiet conditions (0<Kp<2). The standard**
**error of the mean is shown for one example component. Solar maximum (2001-2003) a) zonal, b) meridional average**
**winds. Solar minimum (2005-2007) c) zonal, d) meridional average winds. North and East are purple lines, while**
**South and West are light blue**.



*Fig 6 FPI winter average wind components at KEOPS for geomagnetically quiet conditions (0<Kp<2). The standard error of the mean is shown for one example component.*

*Solar maximum (2001-2003)*
a) *Zonal average winds*
b) *Meridional average winds*

*Fig 6 FPI winter average wind components at KEOPS for geomagnetically quiet conditions (0<Kp<2). The standard error of the mean is shown for one example component.*

*Solar minimum (2005-2007)*
c) *Zonal average winds*
d) *Meridional average winds*





Figure 6 shows the KEOPS FPI winds for clear nights during winter (DOY 300-65) geomagnetically quiet
conditions ($0 \leq Kp < 2o$). The general diurnal trends are similar for both solar maximum (Figs 6a,b) and
minimum (Figs 6c,d). The meridional winds show antisunward flow that is predominantly driven by the
pressure gradient from dayside EUV heating, resulting in fairly weak southward winds reaching a maximum
value of nearly 100 ms$^{-1}$. The standard errors of the mean are around ±10-15 ms$^{-1}$. The zonal winds are eastward
before 1800UT, reaching a maximum speed of a few 10s ms$^{-1}$. After 1800UT the zonal winds turn westward for
a few hours and back eastwards around 2100UT. Between 2100-0300 UT the zonal winds reach their maximum
speed of up to 80 ms$^{-1}$ before turning westward again. The zonal winds are more variable, and their standard
errors of the mean are larger than for the meridional winds, at around ±20-30 ms$^{-1}$.
The few hours of westward flowing zonal winds during 1800-2100 UT are particularly interesting (Figs 6a and
6c). The westward flow indicates that the winds are briefly under the influence of the clockwise dusk cell of
ionospheric convection. Through collisions between the ions and neutral gas, momentum is transferred to the
neutrals, which diverts them from the direction of the pressure gradient driven anti-sunward/eastward flow. The
action of the centrifugal force balancing the Coriolis force keeps the winds entrained in the cell (Fuller-Rowell
and Rees, 1984). As the KEOPS site passes under the region of the Harang Discontinuity (Harang, 1946), the
FPI West zonal winds turn back to eastward about 40 mins after the FPI East zonal winds. This is because the
KEOPS FPI East observing volume is a horizontal distance of 480 km away from the FPI West volume (note
that the distance between the viewing volumes depends on the altitude of the 630 nm emission). However, note
that at the latitude of KEOPS, the time taken for the Earth to rotate through a distance of 480 km is 46 mins. The
difference between 40 min and 46 min is partly due to the difference in magnetic latitude. It is also due to the
Harang Discontinuity being dependent on the IMF $B_y$ orientation, resulting in a smearing out of the MLT
interval.
Figures 6c and 6d show the KEOPS FPI winds for clear nights for the years 2005-2007 during winter (day
numbers 300-65), geomagnetically quiet conditions ($0 \leq Kp < 2o$). These years were during the unusually
extended solar minimum of the last solar cycle when the solar flux levels were extremely low, and observations
of aurora were rare. Consequently the plasma density was smaller, and the thermosphere was more compressed,
resulting in smaller neutral densities at a given height. Under these conditions the ion drag driver is less
efficient, and the pressure gradients, together with the Coriolis and centrifugal forces, play a larger role. Thus,
although the trends are similar to the solar maximum winds; the zonal winds are strongly eastward throughout
the evening sector. There are no westward zonal winds until after 0300 UT, and generally the wind amplitudes
are smaller. The maximum meridional wind is about 80 ms$^{-1}$ southward around 0300 UT. The maximum zonal
winds are seen to the East, and these are around 100 ms$^{-1}$ eastward in the evening sector and start to increase
westwards towards 100 ms$^{-1}$ by 0600 UT. The standard errors of the mean are around ±30 ms$^{-1}$, which are larger
than for solar maximum conditions.
The general trends seen in the CHAMP zonal winds are also seen in the FPI winds. The phases match extremely
well for both (a) Longyearbyen and (b) KEOPS. However, there is a considerable difference in magnitude. The
average ratio of the zonal wind magnitudes (CHAMP /FPI) for Longyearbyen is 1.8, and for KEOPS is 3.3.



**Fig 7 Longyearbyen (Svalbard) winters 2001-2004, 2<Kp<4: average zonal winds measured using CHAMP and FPI, including standard errors of the mean. These are compared with FPI winds in 1980 and the HWM87 and HWM90 model winds from Hedin et al (1991).**

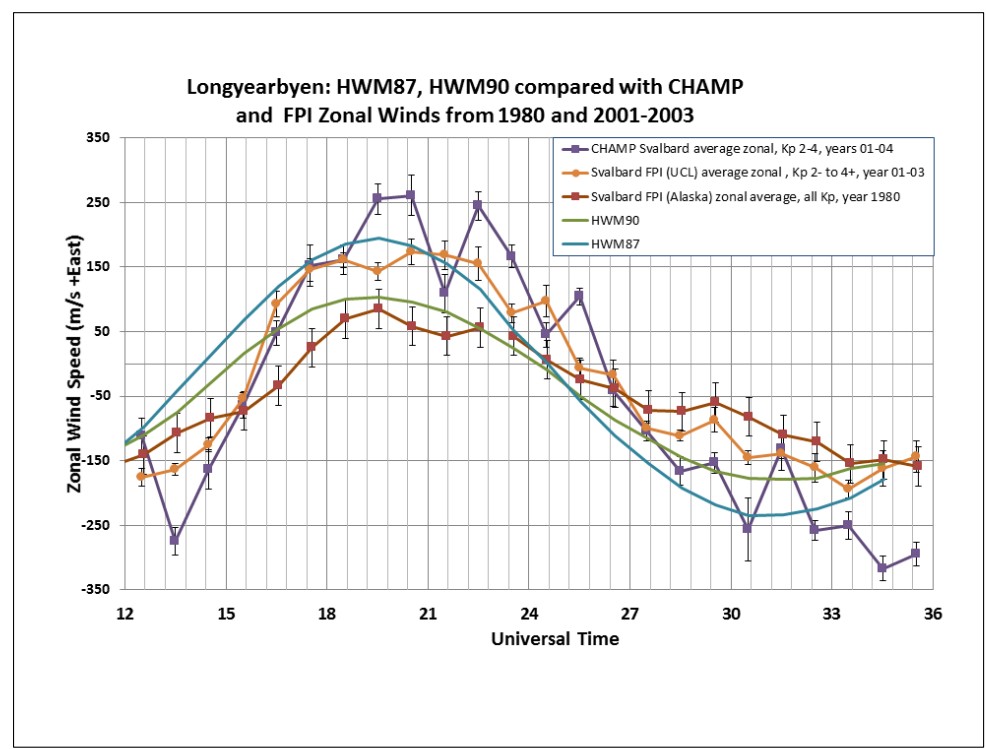

Figure 7 is a direct comparison from several different sources of Longyearbyen zonal winds. These represent moderately active conditions ($2 < Kp < 4$) during winter months (November to January) for solar maximum years that are 20 years apart. CHAMP cross-track winds were collected during the years 2001-2004, while the UCL Longyearbyen FPI observations were during 2001-2003 (a failure occurred in the rotating mirror mechanism in late December 2003). The CHAMP data are the averages of the ascending and descending orbits. Values taken from Figure 9 in the paper by Hedin et al. (1991) are also plotted. These values are measurements from the University of Alaska FPI collected in 1980, 1981 and 1983; and the HWM87 and HWM90 model. The U.Alaska FPI winds are an average of the East and West look directions, justified by the assumption of a uniform horizontal wind field. This was a common practice at that time owing to the longer exposure times of the earlier FPIs (6-12 minutes) which used photomultipliers with piezoelectrical scanning of the FPI etalon gap size in order to view the full Free Spectral Range (Deehr et al., 1980). For Figure 7, only the UCL Longyearbyen FPI East direction winds are shown (Figure 8a shows both East and West look directions). The UCL FPIs were amongst the first FPIs to use fixed gap etalons to image the full FSR onto a 2D array of pixels. This allowed shorter exposure times, and a rapid cycle of look directions. During the 1980s and 1990s we used state-of-the-art UCL designed and built Imaging Photon Detectors (McWhirter et al., 1981) and then EMCCDs (Andor iXon 887/885) were installed around 2005 (McWhirter, 2008). The revolution over the last 30 years in



low light detectors has allowed observations with exposure times as little as 10 seconds at auroral latitudes (Ford
et al., 2007).
All sources show generally similar phases, with peak eastward winds in the evening sector, between 18-24UT
and westward winds in the morning sector, between 06-12UT, as expected for anti-sunward flows. Table 2
shows that the AACGM MLT for Longyearbyen is about 2.4 (± 0.8 for the East and West volumes) hours
ahead, so magnetic midnight is approximately at 21.6 UT. The standard errors of the mean are plotted for all
data. The U.Alaska Longyearbyen FPI standard deviations are around ± 150 ms$^{-1}$, which are similar to the UCL
FPI. For the purposes of comparison, a standard error of ±30 ms$^{-1}$ is plotted for the U.Alaska FPI data, similar to
the average UCL FPI standard error. It was noted in the Hedin et al (1991) paper that for both hemispheres, the
average high latitude winds from the FPIs at Sondrestrom, Longyearbyen and College in the northern
hemisphere, and Mawson in the southern hemisphere, showed a systematically smaller diurnal variation than the
DE 2 mass spectrometer data. The HWM87 model was based on satellite data from the DE 2 and AE-E
satellites. Consequently, the addition of the FPI and incoherent scatter radar datasets to the HWM90 database
resulted in a smaller diurnal variation compared with the HWM87 winds. The more recent measurements from
CHAMP and the UCL FPI are clearly systematically different in magnitude, but consistent with the trend
noticed by Hedin et al (1991) for satellite wind measurements to be larger than from ground-based FPIs. The
diurnal amplitude of the UCL zonal winds is about 170 ms$^{-1}$, and for the U.Alaska winds is about 125 ms$^{-1}$. The
CHAMP zonal winds are systematically the largest in magnitude, with a diurnal amplitude of around 300 ms$^{-1}$.
The average of the monthly F10.7 fluxes is 193 for the winter periods Nov-Jan of 1980-81, 1981-82 and 1983-
84, and 170 for the winters of 2001-2003. Yet despite the higher average solar flux in 1980, the UCL FPI zonal
wind magnitudes have a significantly larger amplitude than the U.Alaska zonal winds. Closer inspection of the 3
winter periods of 2001-2003 shows a spike in the average monthly F10.7 for November 2001-January 2002 (i.e.
<F10.7> is 168 for Nov-Jan 2001-2002;  219 for Nov-Jan 2002-2003; and 152 for Nov-Dec 2003) which may
account for the 3 winter average UCL FPI winds being larger than for U.Alaska. The geomagnetic activity
levels are similar, averaging Kp values in the range 3- to 3o for all three winters.



**Fig 8a Comparison of CHAMP and FPI measurements of Longyearbyen (Svalbard) zonal average winds, including**
**standard errors of the mean, for winters 2001-2004. Plus comparison with Alaska 1980 from Hedin et al. (1991). Fig**
**8b Comparison of CHAMP and FPI measurements of KEOPS zonal average winds, including standard errors of the**
**mean, for winters 2001-2004.**

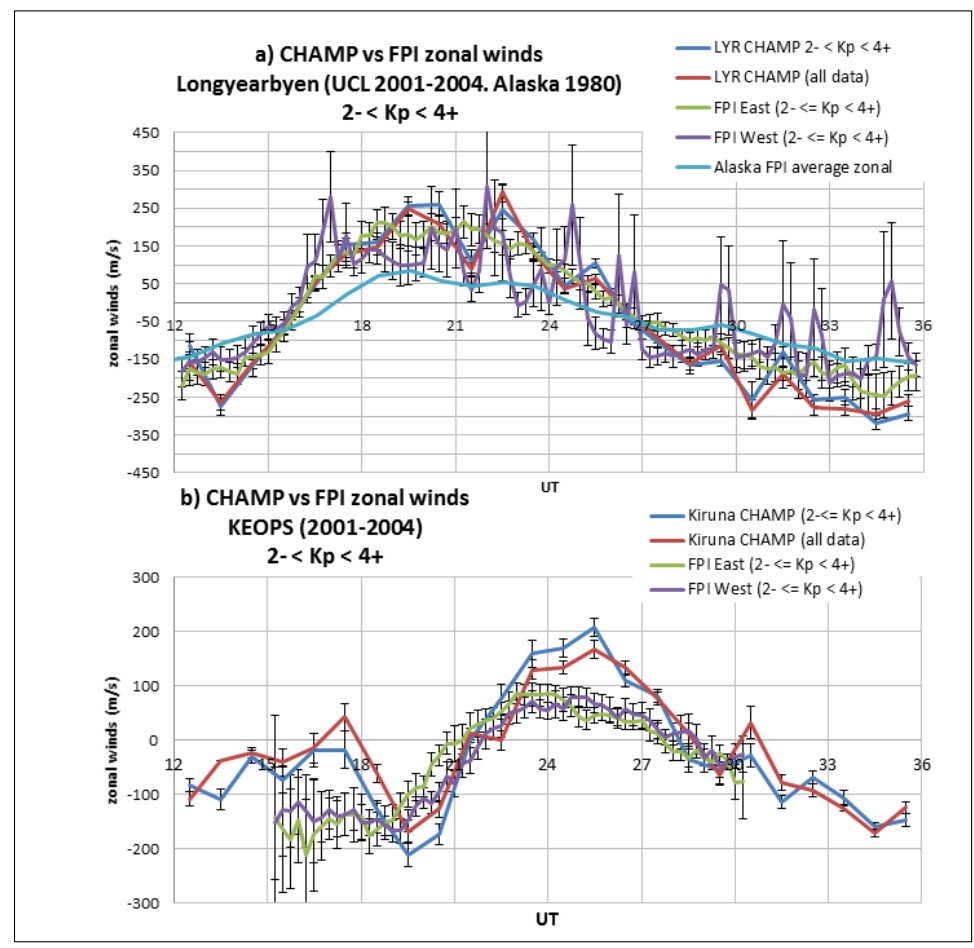


Figure 8a and 8b show a comparison of the CHAMP zonal winds against the UCL FPI winds for the winters of
2001-2004. At Longyearbyen there is 24 hour coverage by the FPI owing to continual darkness between
November-January (Figure 8a). The FPI and CHAMP data are selected for moderately active conditions (2- <
Kp < 4+) as in Figure 7. Here the UCL FPI West look direction is added. Also plotted are CHAMP data for all
activity levels. The Alaska average of East and West look directions from 1980-1983 completes the plot.
Overall there is a very close agreement in phase between the CHAMP and FPI zonal winds. However, there is a
noticeable difference between the UCL FPI East and West look directions. Table 2 shows that the AACGM
magnetic latitude of Longyearbyen West volume is nearly 2° further poleward than the East volume, which



results in weaker emissions and larger error bars. In addition there is a small phase shift owing to the different
magnetic coordinates as discussed further in the next paragraph.
At Kiruna the hours of darkness are between 15-06UT for the period November-January. The UCL FPI average
zonal winds for 2- < Kp < 4+ are shown separately for the East and West look directions in Figure 8b. There is a
smaller difference between these look directions than for the Longyearbyen zonal winds. The evening winds for
moderately active solar maximum conditions are around -150 ms$^{-1}$ (westward), and reach a peak of around 70
ms$^{-1}$ (eastward) in the midnight sector. The AACGM MLT for Kiruna is about 1.9 ($\pm$ 0.3 for the East and West
volumes) hours ahead, so magnetic midnight is approximately at 22.1 UT, which is the time separating the
period of the evening eastward electrojet and the morning westward electrojet in magnetic local time
coordinates. The behaviour of the zonal winds shows strong ion-neutral coupling for these moderately active
conditions, so that there is a semidiurnal variation representative of the twin cell ionospheric convection pattern
at auroral latitudes. This is in addition to the day-night diurnal variation of winds driven by the pressure
gradient.
The phase of the CHAMP zonal winds is in good agreement, but the amplitude is considerably larger. The peak
evening wind reaches -200 ms$^{-1}$ (westward) and 200 ms$^{-1}$ (eastward) by 02 UT. What is particularly interesting
about this comparison is the difference between the CHAMP and FPI winds in the period 15-20 UT. The
CHAMP winds are considerably less westward, and are more similar to FPI average zonal winds for
geomagnetically quieter conditions at solar maximum, as shown in Figure 6a. The large standard error of the
mean during the period 15-20 UT shows how sensitive the winds are to ion drag within the dusk cell.
**Fig 9 Frequency distribution of the ratios of CHAMP/FPI one-hour averaged zonal winds observed in the winter**
**period Nov-Jan for 2001-2004 for Svalbard (red) and Kiruna (blue).**

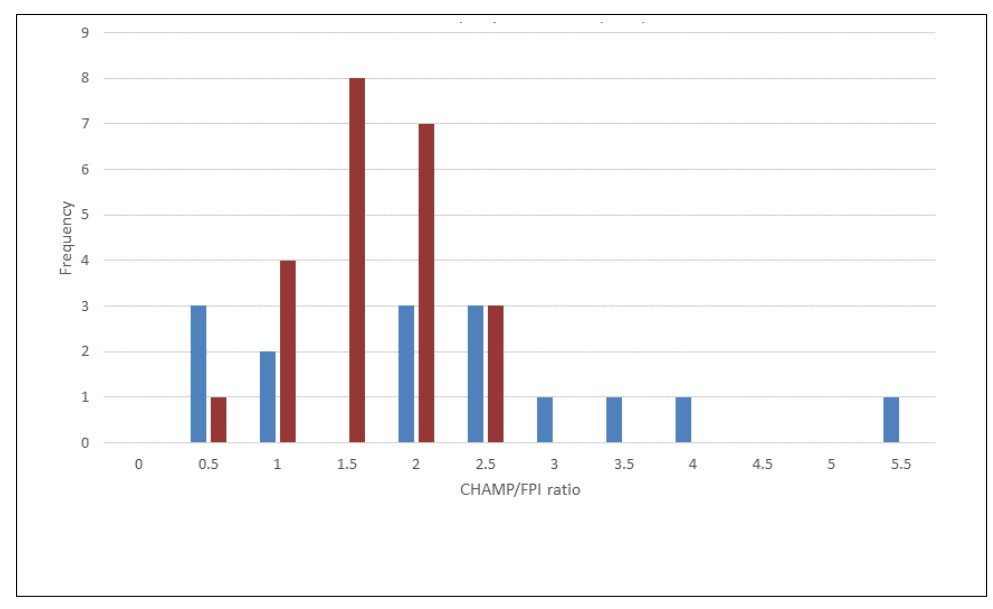




Figure 9 shows a histogram of the frequency distribution of the ratios of CHAMP/FPI average zonal wind
magnitudes observed in the winter period Nov-Jan for 2001-2004 for Kiruna and Longyearbyen. The UCL FPI
zonal winds observed to the East and West have been averaged into 1 hour bins to match the CHAMP averages
of the ascending and descending zonal winds. The Longyearbyen ratios cluster in the range 1.0 - 2.5, while the
Kiruna ratios are more widely spread. Overall there is a general trend for the satellite wind magnitudes to be
larger by a factor of 1.5 - 2.0.
**6. Discussion**
We have shown that there is a similar phase, but a considerable difference between the average zonal wind
magnitudes measured by the CHAMP satellite and the ground-based FPIs for a polar cap and auroral site. Our
premise is that the large viscosity of the upper thermosphere should minimise any vertical structure in the winds
above around 250 km altitude. The difference in wind magnitudes could have various explanations. It could be
that A) we are mistaken about the vertical structure of winds; or B) that there is a problem with the scaling of
the two methods of measurement; or C) the measurement procedures introduce differences, e.g. in-situ versus
remote integration; comparison of different spatial and/or temporal resolutions. There may be other, unexpected,
reasons for the mainly amplitude differences in the measurements.
With respect to hypothesis A: the CHAMP satellite zonal winds are of a similar magnitude to the original
GOCE satellite winds (Liu et al., 2016 and Visser et al., 2019), and to the UCL CMAT2 model simulations.
However, while the CHAMP satellite altitude was between 350-400 km, the GOCE satellite had an unusually
low altitude around 250 km, which was close to the FPI 630 nm emission peak altitude. The CMAT2 winds are
typical of values from other GCMs, which were largely calibrated against measurements by satellites in the
1970s and 1980s, in particular the DE-2 satellites. Killeen et al (1984) found a good agreement between the FPI
at Longyearbyen (then called the University of Ulster FPI, and subsequently the University of Alaska FPI) for
observations in December 1981. This is because the DE-2 satellite measurements were made using the Wind
and Temperature Spectrometer (WATS), rather than derived from satellite drag measurements. The DE-2
satellite flew from August 1981 to February 1983, which means that the average monthly F10.7 flux included
some of the highest solar flux values of the last 30 years. This may account for why the GCMs have such large
wind values.
With respect to hypothesis B: the satellite drag community are aware of a scaling issue. Defining the drag
coefficient is the largest source of error. Bruinsma et al (2014) had to multiply GOCE densities by a factor of
1.29 to match the real-time High Accuracy Satellite Drag Model (Storz et al., 2005). HASDM uses data
assimilation from the orbits of 75-85 inactive payloads and debris over 200-900 km altitude that are tracked by
the Space Surveillance Network (SSN) and is considered a benchmark by that community. Recently March et al.
(2019), reanalysed thermospheric densities derived from very precise satellite accelerometers and GPS
acceleration using high fidelity satellite geometries. The densities for all the spacecraft surveyed were greater
than those derived using surfaces defined by flat panels; and more consistent with each other. The CHAMP and
GOCE densities were found to be 11% and 9% larger. Although there is no simple link between densities and
winds, this re-scaling of densities gives an indication that it may be necessary to scale winds down for the same
measured acceleration (see section 6.3 and Eq. 6).

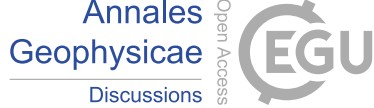

### 6.1 Considering the viscosity of the upper thermosphere

Let us first consider hypothesis A, that the CHAMP and ground-based FPI average zonal measurements are both correct, and that the factor of 1.5-2.0 difference in wind magnitudes is due to the 100-150 km difference in the altitude of the measurements. Conventional fluid dynamics theory predicts that the viscosity is very high in the upper thermosphere owing to the very low particle densities at these altitudes. The viscosity of a fluid determines how resistant it is to shear forces that cause adjacent layers to move at different speeds. Turbulent viscosity dominates the atmosphere below about 100 km, but molecular viscosity dominates the upper atmosphere. The viscosity of the upper thermosphere is very large, and in the CMAT2 model the molecular viscosity $\eta$, is given by Eq. (5) (Harris, 2001) which is based on Dalgarno and Smith (1962) and Banks and Kockarts (1973).

$$\eta = 4.5e^{-5} \times \left( \frac{T}{1000} \right)^{0.71} \tag{5}$$

As a consequence of large viscosity, there is little shear between the different altitude layers above ~200 km for both winds and neutral temperatures (hence the name thermosphere representing an iso-therm behaviour). The issue raised in this paper is that the difference between CHAMP and FPI wind magnitudes is too large to be consistent with the assumption of large viscosity over this range of altitudes.





**Fig 10 CMAT2 zonally averaged zonal winds for 00UT at December solstice 2008 (solar minimum conditions) to**
**demonstrate the effect of drastically reducing the molecular viscosity in order to raise the altitude where winds**
**become independent of altitude. The left panel shows isobars for a standard simulation, while the right panel**
**represents a simulation where the molecular viscosity is 100 times smaller.**



Figure 10 shows two versions of the CMAT2 zonally averaged zonal winds for 00 UT for December solstice
2008 (solar minimum conditions). These are latitude-height plots, where the height is from 15 km to 300 km.
From about 250 km the contour lines become near vertical because the large molecular viscosity of the upper
thermosphere minimises the shear in the winds. The left hand plot is the standard run using standard values of
molecular viscosity. The right hand plot shows the contours for a simulation where the molecular viscosity has
been reduced by a factor of 100. The variation of the molecular viscosity with respect to temperature (and
consequently height for our purposes) has been tested theoretically and experimentally by Dalgarno and Smith
(1962), and the factor of 100 is an unrealistic extreme used to test the model. The consequence is that the height
at which contours become vertical is raised to closer to 280 km. This is a small difference and certainly does not
account for the apparent vertical gradient indicated by the difference between the CHAMP and FPI zonal winds.
Song et al. (2009) studied the local response of the ionosphere and thermosphere to changes of the
magnetospheric convection at polar latitudes on the basis of a 1-D three-fluid model approach. It includes ions,
electrons, and neutral particles and their collisions within the polar cap, hence describing the coupled processes
between the magnetosphere, ionosphere, and thermosphere in the vertical direction along the magnetic flux
tubes. In this self-consistent 1-D solution, the neutral wind speed is obtained as a function of height. The model
describes the dynamic response of the ionosphere/thermosphere after relatively rapid changes of the
magnetospheric convection within about 10-20 Alfvèn wave travel times (or 15-30 minutes) within the
magnetosphere between conjugated hemispheres, or between the ionosphere and the magnetopause, until the
system reaches its steady state again. It is shown that the fastest acceleration of the neutrals occurs near 350 km
in the F layer, where the effective neutral-ion collision frequency maximizes (see Figs. 6 and 7 of Song et al.,
2009). Considering the dynamic character of frequent changes of the IMF and the magnetospheric convection,
the stronger accelerations at F2 layer heights could result in temporary vertical neutral wind gradients. However,
the 1-D model approach neglects forces due to neutral pressure and effective viscosity in the 3-D continuum of
the upper thermosphere (Song et al., 2009). To describe correctly the long-range coupling on time scales from
longer than few seconds to less than 30 min, the inductive effect (Faraday's law) as well as the dynamic effect
of the neutrals, in particular (acceleration terms), need to be considered (Song and Vasyliunas, 2013). This poses
a challenge for future modelling efforts of the M-I-T system.
Recently Vadas and Crowley (2017) published results from observations of 10 Travelling Ionospheric
Disturbances at ~283 km altitude, observed in 2007 with the TIDDBIT ionospheric sounder near Wallops
Island, USA. They used ray tracing on the TIDs and simultaneously measured a peak in the neutral wind at ~325
km altitude using a sounding rocket. They found a serious discrepancy between where the gravity waves were
predicted to dump energy using conventional dissipative theory, and the observations from TIDDBIT and the
rocket. Conventional theory predicted that all the gravity waves should have dispersed at a scale height below
the rocket measurement. Consequently they have proposed that the molecular viscosity should not increase as
rapidly with altitude above 220 km. This may account for some of the difference between the CHAMP and FPI
zonal winds and will need to be tested in future modelling studies.



### 6.2 FPI Doppler shift to wind speed procedure

Hypothesis B is that the FPI and/or CHAMP observations may need to be re-scaled. To start with the FPIs we will look at the calculation of the Doppler shift and then at the height integration procedure of a ground-based FPI.

The calculation of the wind speed requires few assumptions, though the process of fitting the FPI fringes is more complicated (e.g. Makela et al., 2011). The wind speed $u$ is determined from the Doppler shift of the wavelength $\Delta\lambda$ of the moving volume of gas which emits at wavelength $\lambda$, where the free-space wavelength is $\lambda_o$ and the speed of light $c$ (Eq. 7).

$$\lambda = \lambda_o\left(1+\frac{u}{c}\right) = \lambda_o + \Delta\lambda \tag{7}$$

The speed of the volume of gas $u$ is given by Eq. (8), which is proportional to the ratio of the Doppler shift in fringe peak position (in bins) $\Delta x$; and the free spectral range (FSR), $\Delta x_{FSR}$. The FSR is the equivalent wavelength shift to shift a fringe from overlapping one order of the baseline wavelength $\lambda_o$, to the next order. The other terms in Eq. (8) are the refractive index $\mu$ of the medium between the etalon plates and the separation of the plates, $d$ (Hecht and Zajak, 1980).

$$u = \left(\frac{\Delta x}{\Delta x_{FSR}}\right)\left(\frac{c\lambda_o}{2\mu d}\right) \tag{8}$$

The etalon gap is evacuated so $\mu=1$, and the other parameters are known. Thus for example, for an etalon gap $d = 10$ mm, emission $\lambda_o = 630$ nm, free spectral range $\Delta x_{FSR} = 150$ bins, a Doppler shift of 1 bin ($\Delta x = 1$ bin) would represent a wind of 63 ms$^{-1}$.

All the parameters for the scaling of the FPI winds in this equation are known. There is the issue of determining the zero Doppler shift baseline because there is no laboratory source of the excited atomic oxygen. However, the method used to determine the baseline (i.e., using a helium-neon source with the assumption that the vertical component of the wind is negligible) introduces an average systematic offset error of at most 10-20 ms$^{-1}$, which is small compared with horizontal wind magnitudes (Aruliah and Rees, 1995).



**Fig 11 left: height profile of CMAT2 zonal winds at Svalbard. Right: height profile of the red line emission intensity**
**profile from the Vlasov et al (2005) model.**

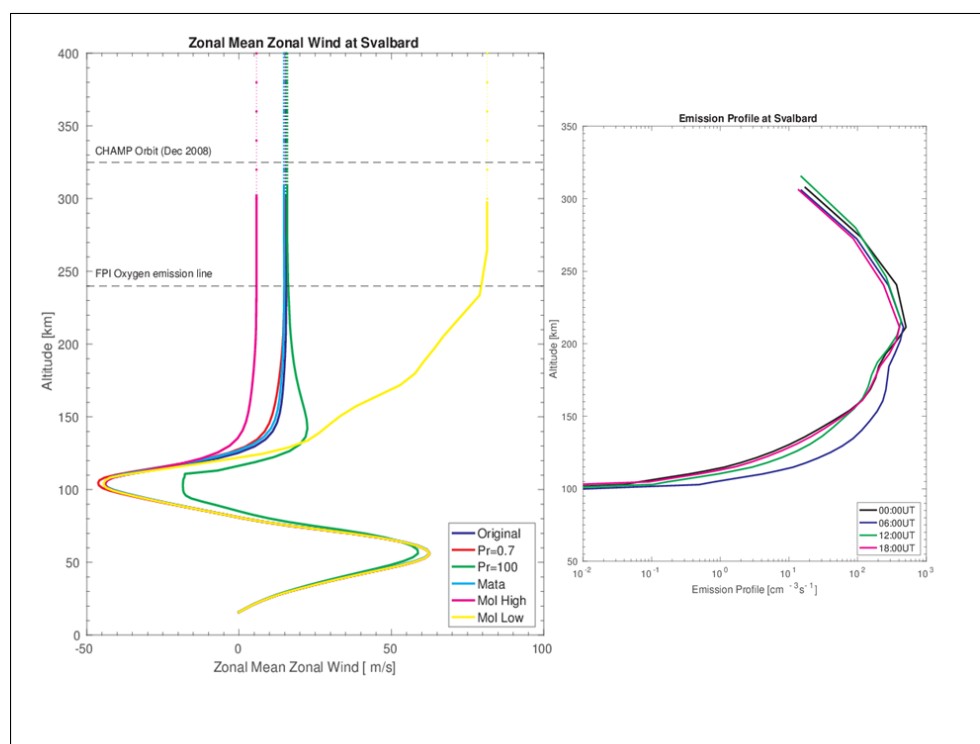

Figure 11 illustrates how ground-based FPIs make measurements of the neutral winds at 240 km altitude. The
left plot shows a height profile of the CMAT2 zonal winds at Longyearbyen. On the right is a height profile of
the 630 nm (red) line emission intensity based on the Vlasov et al. (2005) model. The red line emission at night
is dominated by dissociative recombination of molecular oxygen ($O_2^+$ + e -> O* + O) (e.g. Vlasov et al., 2005).
The ground-based FPI measures the Doppler shift of the gas, height integrated through a volume along the line-
of-sight. This means that all winds at all altitudes contribute to the measurement, but are weighted by the red
line emission intensity. The emission height profile shows a sharp velocity gradient below 200 km, but owing to
quenching of the emission through collisions, there is little emission below 200 km (note the emission intensity
x-axis is a log scale), and therefore a minimal contribution to the height-integrated line-of-sight wind
measurement. Above 200 km the wind magnitudes begin to reach an asymptote. It therefore would be expected
that the satellites and ground-based FPIs should see very similar speeds and phases. The tristatic FPI
experiments by Aruliah et al (2005) and bistatic experiments by Anderson et al (2012) indicated that the winds,
neutral temperatures and 630 nm intensities were closely matched if the geometry assumed an emission altitude
of around 240 km. However, during auroral activity, when there is E-region precipitation, the red line emission
altitude can be lower, perhaps as low as 200 km. This means that the FPI samples lower altitudes. Recently
Gillies et al. (2017) used all-sky imagers to triangulate the peak emission height of the 630 nm emission. They
found that discrete auroral arcs showed a characteristic height of 200km. The effect of particle precipitation in
lowering the emission height was earlier noted by Sica et al. (1986). They illustrated how decreased




thermospheric temperatures measured by a Fabry-Perot spectrometer at College, Alaska, were consistent with
lower MSIS temperatures (Hedin et al., 1977) when weighted by a modelled emission height profile. However,
aurorae are limited to high latitudes and occur infrequently as illustrated by Figure 12, which shows the
frequency distributions of Kp values for the years (top) 2001-2003 representing solar maximum; and (bottom)
2000-2009, i.e., for most of the period of the CHAMP lifetime. Aurora generally occur during active periods
when Kp > 4-5. Thus emission heights of 200 km are the exception rather than the rule.
**Fig 12 Frequency distribution of Kp values. Top: 2001-2003 representing solar maximum. Bottom: 2000-2009**
**covering most of the period of the CHAMP lifetime.**

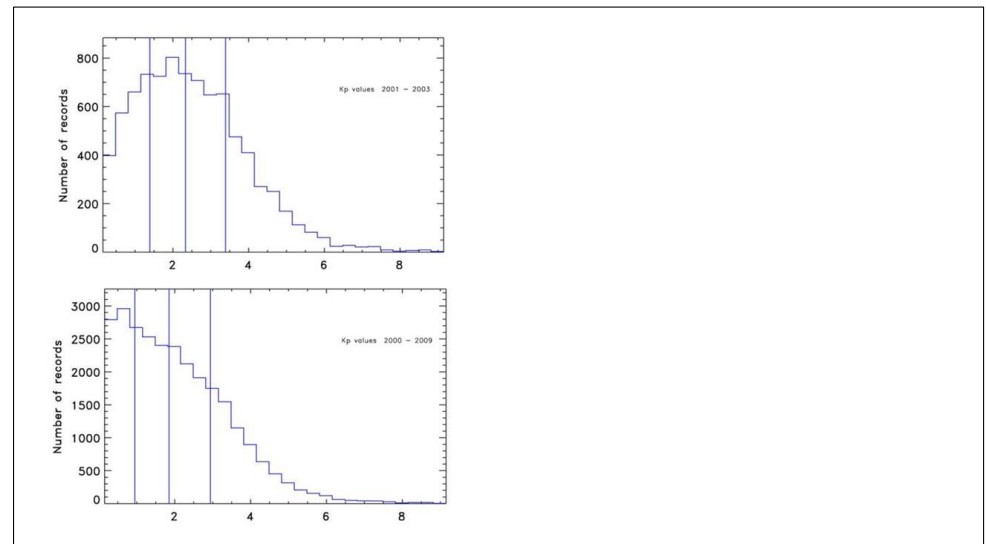





**Fig 13 top: CMAT2 zonally averaged zonal winds for a quiet day on 1ˢᵗ December 2007 at Longyearbyen (left) and**
**Kiruna (right) for the winds at 180, 200 and 240km for comparison with the height integrated winds weighted using**
**an emission intensity profile from the Vlasov et al (2005) model. Bottom: the same for active conditions on 20ᵗʰ March**
**2015.**

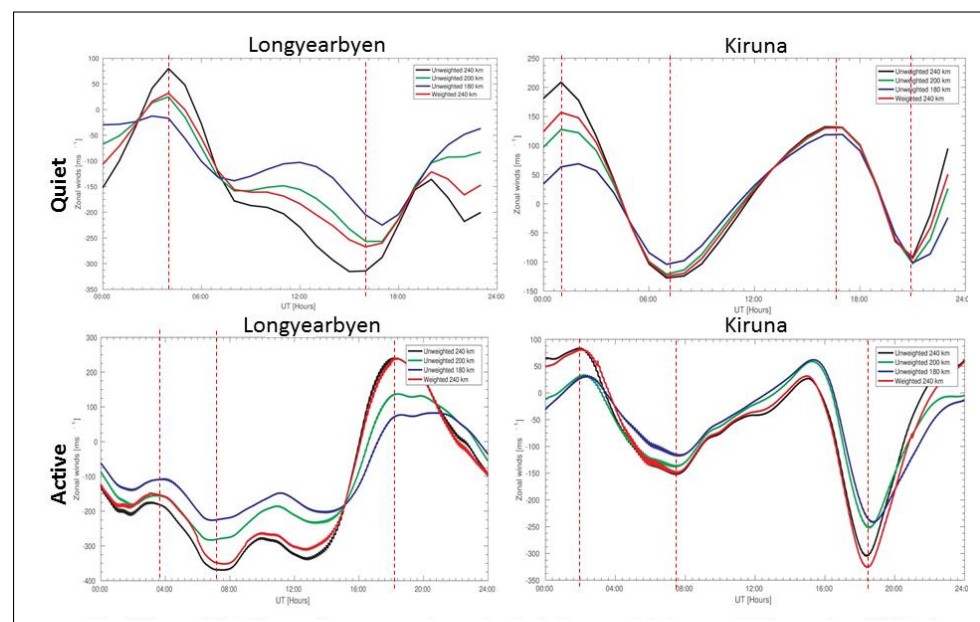


In order to assess by how much the FPI height integration method is underestimating winds, CMAT2 winds at
240 km are compared to a column integration average of CMAT2 winds weighted by the emission intensity
profile. Here the Vlasov et al (2005) model is applied with constants provided by Yiu (2014), and with CMAT2
winds interpolated to 10 km intervals for the integration. Figure 13 compares the CMAT2 zonally averaged
zonal winds at three heights: 180 (blue), 200 (green) and 240 km (black) with height integrated winds (red) for a
quiet day run on the 1ˢᵗ December 2007 (top panels) and an active day run on the 20ᵗʰ March 2015 (bottom
panels) for both Longyearbyen (left column) and Kiruna (right column). Figure 14 outlines the CMAT2 model
global view of the unweighted and weighted winds at 240km for 00, 06, 12 and 18UT.

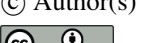



**Fig 14: CMAT2 global zonal winds for a quiet day on 1$^{st}$ December 2007 for the winds at 240km and the height integrated winds weighted using an emission profile from Vlasov et al (2005) model. From top left: 0UT, 6UT. From bottom left: 12UT, 18UT.**

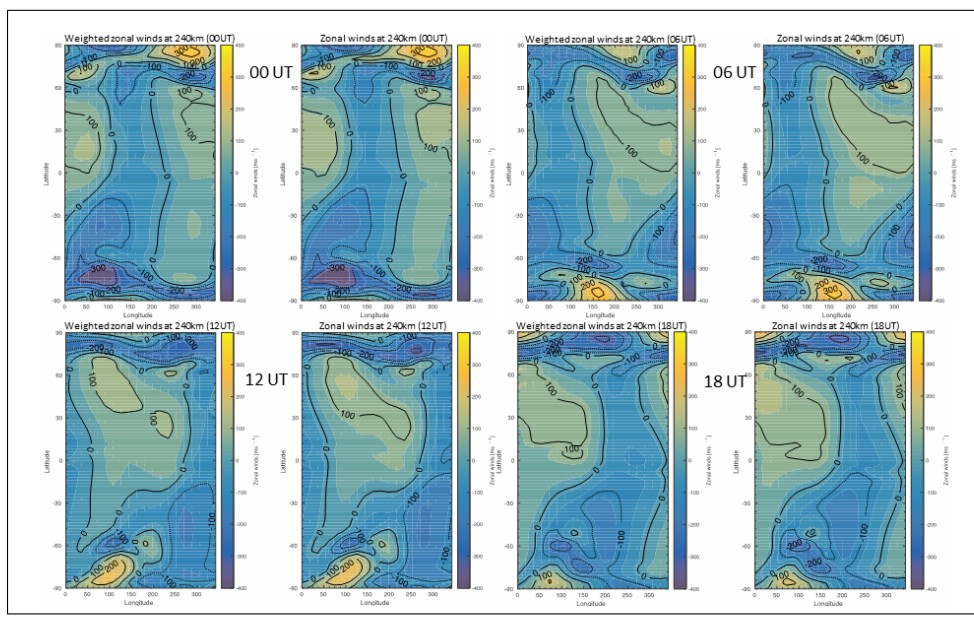

Figure 13 indicates that there are some significant differences between zonally averaged zonal winds with or without height integration. The lower the altitude, the smaller the wind magnitude. There is also a slight change in phase (to aid the eye these are indicated by vertical dashed lines placed at turning points for the weighted winds). This is due to the increased collision frequency at lower altitudes due to greater density, and the consequent shift in balance between pressure gradient and ion drag. When we look at the averaged diurnal variation of the CHAMP and FPI winds, their phases are almost exactly the same. This would not be the case if the FPI was observing winds dominated by Doppler shifts at 190km altitude, where the phase would be different because the pressure gradient increases its dominance at lower altitudes. Comparing the CMAT2 December 2007 model zonally averaged zonal winds at 240 km with the height integrated winds, the most significant difference is for Longyearbyen during quiet conditions, which on calculating overall mean values produces a 19% difference, that during active conditions reduces to a difference of 12%. The reverse seems the case for Kiruna as the mean percentage difference increases from 3% to 14% between the quiet and active days respectively. Note that the wind speed scales are different for each panel. For each time-series there is no simple systematic trend. Figure 14 demonstrates these dissimilarities on a global scale; here the zonal winds appear to be slightly more westward, as the Eastward winds are diminished and the Westward winds enhanced. However, this is not the case for all times of day shown here, and is not visibly affecting the wind distributions to a large extent.



**6.3 CHAMP cross track wind procedure**

Satellites have provided global coverage of accelerometer measurements since 2001, in particular, the CHAMP
satellite (e.g. Schlegel et al., 2005) and GRACE satellites (e.g. Tapley et al., 2004). These measurements of
satellite drag have been converted to measurements of thermospheric density (e.g. Liu et al., 2005) and cross-
track thermospheric wind measurements (e.g. Sutton et al, 2007, Liu et al., 2006, Förster et al., 2008) using Eq.
6, where $\underline{a}$ is the satellite acceleration, $\rho$ the neutral mass density of the air, $C_d$ refers to a dimensionless drag
coefficient, using a constant frontal area $A_{ref}$ of the satellite with mass $m$, and total velocity $V$ relative to the
atmosphere in the ram direction given by the unit vector $\underline{\hat{v}}$.

$$\underline{a} = -\frac{1}{2}\rho\frac{C_d}{m}A_{ref}V^2\underline{\hat{v}} \tag{6}$$

For the first analysis, Liu et al. (2005) explained that they used a fixed drag coefficient value of $C_d = 2.2$. This is
a de-facto standard value used for compact satellite orbit computations since the 1960s (e.g. Cook, 1965). This
value was adopted by Jacchia when constructing his thermosphere density model, based on physical drag
modelling of spherical satellites (Jacchia and Slowey, 1972). The drag coefficient is acknowledged to be very
difficult to quantify, as is discussed extensively by, for example, Moe et al. (1995). The importance of the value
of $C_d$ is acknowledged by Liu et al (2006) and others who use the data, since it affects the scaling of the density
and wind calculations. However, their interest was in the relative density and wind structures, rather than
absolute values. Since then the analysis has been refined considerably by taking into account lift, sideways, as
well as drag forces on the satellite, resulting in smaller wind magnitudes as described by Doornbos et al. (2010).
The GOCE satellite winds are closer in magnitude to ground-based FPI measurements (Dhadly et al., 2017),
though still systematically larger in magnitude, where the difference has been found to increase with latitude.
The systematic residual line-of-sight GOCE wind varied between 20 ms$^{-1}$ at 50° MLAT to a maximum 150 ms$^{-1}$
at 85° MLAT (see Figure 2 from Dhadly et al., 2018).

Another consideration is that CHAMP measures the cross-track wind component (Figures 1 and 2) which
deviates from the pure zonal direction as measured by the FPIs (Figures 3 and 4). The geometry can be critical,
in particular for the high-latitude Longyearbyen FPI, because the cross-track deviates from the zonal direction
by about 13.5° in each direction respectively for the ascending and descending orbits. The meridional wind
component at these high latitudes is much larger than the zonal one, so that the larger CHAMP measurements at
this FPI could also (at least partially) be due to an "admixture" of the meridional wind component and the zonal
wind. This has been discussed in section 5.1 to account for the difference between the average zonal winds
measured during the ascending and descending orbits.

**6.4 Comparison with EISCAT radar ion velocities**

Finally, a very important consideration is how the average winds compare with ion velocities. At high latitudes
the ion velocities are generally larger than the neutral winds owing to the ExB drift driven by the
magnetospheric electric field. Davies et al (1995) provided a statistical analysis of E- and F- region ion





velocities observed on 20 March 1996 in order to compare measurements by the EISCAT incoherent scatter
radars and the CUTLASS coherent scatter radar. The scatter plot of ion velocities from this study (their Figure
5) indicated a cluster of values in the range of a few hundred ms$^{-1}$, with only a small fraction of measurements
greater than 500 ms$^{-1}$.
Fiori et al (2016) compared ion velocities measured by the Electric Field Instrument on Swarm with the CS10
statistical ionospheric convection model by Cousins and Shepherd (2010) which is based on 8 years of data
(1998-2005) collected by 16 SuperDARN coherent scatter radars. The climatology represented by the CS10
model in Fiori et al's Figure 3a indicates speeds in the few hundreds of ms$^{-1}$, while the instantaneous values
along the Swarm satellite pass in their Fig 3d show horizontal velocities over a 1000 ms$^{-1}$, which probably
indicates the dynamic behaviour in the auroral regions.
Aruliah et al. (1996) presented the seasonal and solar cycle variation of hourly averaged ion velocities from 300
days of EISCAT Tromsø UHF radar measurements between 1984-1990. The tristatic EISCAT radar
observations for an altitude of 275 km were collected from Common Programmes 1, 2 and 3, at time resolutions
of 2-3 mins, with full 24 hours coverage. The ion velocities for December solstice periods were up to 100-200
ms$^{-1}$, and the largest average ion velocities were around 300 ms$^{-1}$ during the March equinox period at solar
maximum. Aruliah et al. (2005) later reported observations of a common volume using a configuration of
tristatic FPI observations of the thermospheric winds and temperatures co-located with tristatic EISCAT radar
measurements of ionospheric parameters at 250 km altitude. The observations showed that the neutral winds
were on average around 50% of the magnitude of the 15 min average ion velocities.
Griffin et al. (2004) determined seasonal and solar cycle climatologies of meridional winds at Kiruna using FPI
Doppler shifts, and derived from field-aligned ion velocities (Salah and Holt, 1974), which were compared with
physical (CTIM, Fuller-Rowell et al., 1988) and empirical models (HWM, Hedin et al, 1988; MWM, Miller et
al., 1997). The climatologies all showed meridional winds up to ~250 ms$^{-1}$. Although this method does not give
the zonal wind magnitude, it gives some indication of typical magnitudes owing to the diurnal variation of
winds seen by a single site as the Earth rotates.
Förster et al. (2008) presented a statistical comparison of observed averaged neutral wind velocities within the
polar cap (magnetic latitudes > 80°) for the year 2003 showing the dependence on the IMF orientation based on
statistical analyses of CHAMP accelerometer data with average ion drift estimates for the same time interval
and IMF conditions based on EDI Cluster measurements. These comparisons were done for both the Northern
and the Southern Hemisphere separately in their Tables 1 and 2, respectively. Depending on the IMF clock
angle orientation, the ratio between average neutral wind magnitudes and average ion drift speeds varies
between about 60% and around 100%. Interestingly, there is a characteristic interhemispheric difference with
respect to the IMF orientation and slightly larger ion drift velocities on average in the Northern Hemisphere (cf.
Förster and Cnossen, 2013; Förster et al., 2017), but the overall average amounts to a ratio of about 0.90 to 0.95
for, note well, within the polar cap region > 80° magnetic only. The FPI in Longyearbyen at 75.4° N (see Table
2) comes closest to this region. Ion drag is the dominating forcing term here for the neutral gas, while near the
auroral ring, where the KEOPS FPI station in Kiruna at 65.1° N is located, the balance between the different
forces, in particular pressure gradient terms, Coriolis and centrifugal forces, and ion drag play a role. There the



ratio between the average neutral wind and ion drag magnitudes is certainly smaller, corresponding to the
EISCAT observations cited above.

**7 Conclusions**

A comparison is presented here of thermospheric zonal winds measured by the CHAMP satellite in the altitude
region 350-400 km, and by ground-based FPIs, at Kiruna and Longyearbyen, measured at about 240 km
altitude. The satellite accelerometer measurements are used to derive cross-track winds, while the FPIs use the
Doppler shift of the 630 nm emission. The satellite measurements are collected for a region within 2° of the FPI
sites, which is within the field of view of the FPI East and West look directions. The phases of the winds agree
very well, but the CHAMP average zonal winds are a factor 1.5-2.0 larger than the FPI average zonal winds.
The factor is not simple. In particular there is a difference in the factor for the auroral site and the polar cap site,
so it appears that the factor is dependent on location, possibly latitude.
The UCL Longyearbyen FPI winds are consistent with FPI measurements made 20 years previously by the
University of Alaska using a different FPI and detector (photometer in 1980, EMCCD in 2001). Earlier studies
of average ion velocities from the EISCAT Tromsø UHF radar compared with the UCL FPI at KEOPS indicate
that in the auroral zone the average ion velocities are about twice the average neutral wind speeds (Aruliah et al.,
1996 and 2005). However, the CHAMP average KEOPS zonal winds presented here have magnitudes similar to
the average _ion_ velocities of the December solstice values presented by Aruliah et al (1996). It is important to
determine the absolute wind values since the difference between the ion and neutral winds determine the amount
of Joule heating of the thermosphere.
Satellites play a crucial role in upper atmosphere research by filling in the extensive gaps between ground-based
observations. Satellites provide 3-dimensional coverage at high spatial resolution, in addition to high temporal
resolution. Meanwhile, ground-based instruments are sparse, land-based, and not always operational on a 24/7
basis owing to operational costs (e.g. incoherent scatter radars) or observing constraints (e.g. only night-time
and clear sky observations for optical instruments). Having uncovered this discrepancy between ground-based
FPI optical measurements and satellite drag measurements of winds, it is imperative to determine if it is a real
altitude dependence, or if some re-scaling of winds, is necessary for winds determined from either, or both, of
FPI height-integrated Doppler shifts or satellite drag measurements. Both will affect our current modelling of
the upper atmosphere; or whether we need to rethink the procedure of comparing different spatial and temporal
resolutions of in-situ satellite versus remote ground-based FPI measurements and the geometry of cross-track
winds at high latitude.

*Author contributions.* This paper is the result of many years of collaboration between AA and MF after noticing
the significant difference between FPI and CHAMP winds. AA provided the FPI data, MF provided the
CHAMP data. RH provided the model simulations, IM provided technical support for the FPIs, and ED
provided expertise on converting accelerometer data to winds.





*Competing interests.* The authors declare that they have no conflict of interest

*Acknowledgements*. We thank the staff at the Kjell Henriksen Observatory and ESRANGE for hosting the FPIs
and their generous on-site assistance. Support for the FPI operations have come from the European Office of
Aerospace Research and Development (grant FA9550-17-1-0019). There has been NERC support of ALA
(grants NE/P001556/1 and NE/N004051/1). The CHAMP mission is sponsored by the Space Agency of the
German Aerospace Center (DLR) through funds of the Federal Ministry of Economics and Technology,
following a decision of the German Federal Parliament (grant code 50EE0944). The data retrieval and operation
of the CHAMP satellite by the German Space Operations Center (GSOC) of DLR is acknowledged. EISCAT is
an international association supported by research organisations in China (CRIRP), Finland (SA), France
(CNRS, till end 2006), Germany (DFG), Japan (NIPR and STEL), Norway (NFR), Sweden (VR), and the
United Kingdom (STFC). We also acknowledge support from the International Space Science Institute for
sponsoring meetings of the international team #308 on 'M-I-T Coupling: Differences and similarities between
the two hemispheres', which helped this collaboration (http://www.issibern.ch/teams/twohemispheres/).

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
