# Peer review of "Comparing high-latitude thermospheric winds from FPI and CHAMP accelerometer measurements"

_Annales Geophysicae, 2019_

## Referee Comment (RC1) · Anasuya Aruliah et al. · 9 May 2019

Dear Editor, I have reviewed the paper by Aruliah et al. titled "Comparing high-latitude thermospheric winds from FPI and CHAMP accelerometer measurements." The paper describes polar observations from FPIs and the CHAMP satellite. Observations from different instruments are compared and discrepancies discussed. It is important to the community to understand the differences between the FPI and CHAMP observed thermospheric winds. CHAMP winds are known to be larger than the FPI measurements. While the cause for such discrepancy may be unknown at the moment, at least, we should know how the two sets data are different. Hence, the paper is very important and should be considered for publication. It can be a good reference for future users of the two data sets. The paper in the current form, however, has some significant issues

need to be addressed.

1. The paper somehow lost focus. It got into too many sub-topics: FPI (old) -FPI (new) differences, HWM87-HWM90 differences. I think for the purpose of understanding the CHAMP and FPI difference, we should avoid using very old data. If the topic were the long-term trend, then we should examine long data string. The value of this paper is on the CHAMP and FPI comparison.

2. The section L67-89 has some major issues. The thermospheric dynamics is governed by the momentum, energy, and continuity equations. It cannot be expressed in the formulas listed in the section. CHAMP's larger winds do not necessarily lead to larger temperatures. Smaller FPI winds are not always connected to the lower temperatures. We cannot use temperature to verify wind values.

3. I am lost in the section L279 – L316. I understand that the CHAMP cross-track winds are not aligned with the eastward direction on the ground. The angle is different for the ascending and descending nodes. The angles are +/- 7.2 deg and +/-13.3 deg for KEOPS and Longyearbyen (ascending and descending). So the obvious thing to do is to compare separately the ascending and descending nodes. Use the ground based meridional and zonal components to compute the wind value along the direction of CHAMP cross-track measurement (mostly to add the contribution from the meridional winds). I don't understand why that was not done. I do not see an argument to let me believe that I can ignore the viewing angle difference between the ground based zonal wind and CHAMP cross track winds. So I suggest the authors use the ground based FPI data to form the wind values along the CHAMP cross-track direction to do a direct comparison. Or alternatively, using the CHAMP ascending and descending node measurements to remove the contribution from the meridional wind and compute the zonal wind. Hopefully, that would give you better comparison between the FPI and CHAMP observations.

4. Figure 7 should be the focus of the paper. I really don't see much value having two

HWM model runs results shown here. You only need one. The FPI (Alaska) probably does not add much and more likes a distraction. The old and new FPI comparison should be discussed in a different paper, you can have inter-annual variations here.

5. The solar minimum data are not very useful since there are no CHAMP observations.

6. I think the conclusion should be clearer that the CHAMP winds are overestimated. As the paper points out that the CHAMP winds are almost the same magnitude as ion drift at Kiruna, which is incorrect.

7. While Aruhiah et al. 2005 reference is listed for the instrument information. It will be a great help to give a short paragraph on the two FPIs (gaps, aperture size, detector, from which year to which years) given that instrument upgraded over the years.

Minor

1. In the comparison section, the FPI (Alaska) was used, but there is no mention of it in the abstract and earlier instrument description. It should be added to the abstract, if it is to be used. Personally, I think the data should be dropped.

2. Many the figures have very low resolution and are difficult to read.

3. Figure 5, the FPI from one direction has error bars and the other does not. Why?

4. L 444 to LL448. 'During the 1980s and 1990s we used state-of-the-art UCL designed and built Imaging Photon Detectors (McWhirter et al., 1981) and then EMCCDs (Andor iXon 887/885) were installed around 2005 (McWhirter, 2008). The revolution over the last 30 years in . . .' Was the UCL FPI running with the imaging photon detector from 2001 to 2003, it is not clear in the paper. That is why I ask for a more detailed instrument description to be added.

5. I do not see Harris 2001 paper (L554) in the reference. Or the reference date is wrong? It is not 2017, should it be 2001?

6. Equ. 5 is wrong. It does not match the two references.

7. There should be some discussion on hypothesis C (L520).

8. Why is the HWM model run not included in Figure 8 comparison for Kiruna?

---

## Referee Comment (RC2) · Anonymous Referee #2 · 31 May 2019

This paper statistically compares upper thermospheric F-region winds measured by two high-latitude ground-based Fabry-Perot Interferometers (one located near Kiruna and other at Longyearbyen) and derived from in-situ accelerometer measurements on-board the CHAMP satellite. One of the ground-based stations is located in the auroral zone whereas the other one is in the polar cap. Results show that CHAMP winds are systematically 1.5-2 times larger than FPI winds. Further, the authors utilize the existing modeling tools for exploring the various possible reasons responsible for these systematic discrepancies in winds obtained from in-situ and optical techniques. Overall, this study can serve as an important reference for data users of these instruments.

[Figure]

In my view, the manuscript is loosely written. There is some repetitiveness of some of the text and the manuscript could be streamlined quite a bit. I would strongly recommend the authors to make clear, elaborate, and explain the following parts:

1. Please explain the purpose of having fist four figures (Figures 1-4). I think they are irrelevant and can be dropped without impacting the focus of the paper. Instead, it would help focusing this study on the core topic - FPI and CHAMP wind comparison.

2. Line 17: should be kinematic viscosity instead of viscosity?

3. Line 25: +-2 degrees in latitude, longitude, or both? Please explain.

4. Line 148: In Table 1 (column 4 and row 2), you mean 1.860 UT?

5. Line 172: Emmert 2006a reference is not valid here because it is a climatological study.

6. Lines 299-317: The simplest and most direct way to compare CHAMP and ground station winds would be to project ground station winds along the CHAMP cross track winds; it is doable because both the zonal and meridional winds exist for ground station FPIs.

7. Figure 5 and 6: Please keep the figure titles consistent. Subfigures a/b titles are not consistent with c/d titles: one shows Kp index in title and others not. In addition, please keep consistency when using plus or minus symbols in Kp values. For example, sometimes the manuscript uses Kp<2 and the other times Kp<2- [[or Kp<2o (line 367, 413, etc.) which may be a typo]]. Kp<2o is also present in Figures 6a and b. Moreover, I would suggest using an actual math symbol (ïĆč) instead of <=.

8. Lines 424-426 are referred to which figure/figures?

9. Figure 7:

- This comparison is done for Kp 2-4, whereas earlier figures and discussion was focused on Kp 0-2. Same is true for Figure 8. Please explain the reason for this gear shift.

- Please explain why HWM87 and HWM90 were used instead of HWM14? HWM14 is the latest version of this empirical wind model.

10. Figure 9: In addition to this figure, a plot showing CHAMP/FPI ratio as a function of UT or LT would be really helpful.

11. Lines 518-522: The major source of discrepancies could be the assumptions used when applying different wind extraction schemes as they can fail under different conditions.

12. Line 556: Please verify the viscosity expression.

13. Lines 715-722: Project FPI wind vector along the CHAMP cross track wind component.

14. Section 6.4: I did not get the motive of adding this section. So, please state explicitly the contribution of this section in this investigation.

---

## Author Response (AR1)

[revised manuscript text omitted]

$$\delta E = \frac{1}{2}\bar{\rho}U_{sat}^2 + \bar{\rho}C_p \delta T_{sat} = \frac{1}{2}\bar{\rho}U_{FPI}^2 + \bar{\rho}C_p \delta T_{FPI} \tag{2}$$

Consider if the measurements from the satellite and FPI are such that $U_{sat} = 2U_{FPI}$, i.e. either the satellite or the FPI (or both) are wrongly calibrated, then substituting for $U_{sat}$ in Eq. (2), and rearranging both equations, leads to Eq. (3).

$$4\delta T_{FPI} - \delta T_{sat} = \frac{3\delta E}{\rho C_p} \tag{3}$$

Thus Eq. (3) demonstrates that for positive $\delta E$ (i.e. heating), the inferred satellite temperatures are appear to be larger than the FPI temperatures (e.g. if $\delta E \approx 0$, then $\delta T_{sat} \approx 4\delta T_{FPI}$). In other words, by applying the satellite wind measurements, the implication isy that more energy is put into heating the gas, and less into accelerating the gas., Meanwhile applying the FPI wind measurements would indicate the reverse. This would result in a mismatch between modelled and observed temperature changes. The FPI can measure temperatures to test this, as will be done in a future study. The temperature discrepancy would also have a knock on effect on the calculation of density $\rho$ of the gas as determined by the satellites, or by ground based FPIs, since $\delta\rho = nk_B\delta T$, where $k_B$ is the Boltzmann constant and $n$ is the number density of the gas particles. Note that this argument for a point localised measurement is oversimplified. The purpose is to highlight the repercussions of overestimating (or underestimating) the neutral wind on the division of energy between heating and acceleration of the neutral gas. The conservation of momentum, energy, and continuity must be satisfied, and oOwing to the high molecular 
[revised manuscript text omitted]

[Figure]

a) All data – ascending (blue) and descending (red) averages b) Summer (May-Aug) – ascending (blue) and descending (red) averages c) Winter (end Oct-early Mar) – ascending (blue) and descending (red) averages

**Figure 54: CHAMP observations over KEOPS during solar minimum 2005-2007. a) All data – ascending (blue) and**
**descending (red) averages; b) Summer (May-Aug) and c) Winter (end Oct-early Mar).**

[Figure]

a) All data – ascending (blue) and descending (red) averages b) Summer (May-Aug) – ascending (blue) and descending (red) averages c) Winter (end Oct-early Mar) – ascending (blue) and descending (red) averages

[revised manuscript text omitted]

Fig 8 Longyearbyen: CHAMP versus HWM93
and FPI crosswind vector component
1-hour averages for 2001-2003 and 2- ≤ $Kp$ < 4+

zonal winds measured using CHAMP and FPI, including standard errors of the mean. These are compared with FPI winds in 1980 and the HWM87 and HWM90 model winds from Hedin et al (1991).

Figure 87 is a direct comparison between CHAMP cross-track winds at Longyearbyen from and several different sources. The winds represent moderately active conditions (2- ≤ Kp < 4+) during winter months (November to JanuaryDOY 300-65) for solar maximum years. The HMW93 model is setconditions are for the 31st December atwith 
[revised manuscript text omitted]

[Figure]

[Figure]

Figure 10a) ratio of absolute CHAMP/FPI combined East-West and +/- α cross-track winds for solar max, 2- ≤ $Kp$ < 4+

[Figure]

Figure 10b) UT dependence of ratio of absolute CHAMP/FPI combined East-West and +/- α cross-track winds for solar max, 2- ≤ $Kp$ < 4+

Fig 9 Frequency distribution of the ratios of CHAMP/FPI one-hour averaged zonal winds observed in the winter
period Nov-Jan for 2001-2004solar max under moderately active conditions (2- ≤ $Kp$ < 4+) for Svalbard
Longyearbyen (redblue) and Kiruna (bluered). Fig 9a) shows UT dependence of ratios, and Fig 9b) shows frequency
distribution of ratios.

[Figure]

Figure 10a) ratio of absolute CHAMP/FPI combined East-West and +/- alpha cross-track winds for solar max, 2- ≤ $Kp$ < 4+

[Figure]

Fig 10b) UT dependence of ratio of average absolute CHAMP/FPI combined East-West and +/- alpha cross-track winds for solar max, 2- ≤ $Kp$ < 4+

Figure 9a) UT dependence of average CHAMP/KEOPS magnitudes

[revised manuscript text omitted]

2014.

**21st August 2019**

**We would like to thank the reviewers for their time taken to read and feedback very helpful**
**advice and comments on our paper. Our responses are below.**

We have added an extra figure (Figure 3) to illustrate the geometry of the CHAMP ascending and

[Figure]

Figure 3: Geometry illustrating the projection of FPI look direction wind components onto the CHAMP cross-track direction for the ascending and descending tracks.

descending orbits, and the projection of the FPI wind vectors onto the CHAMP cross-track direction.

This is in response to both referees requesting that we do this projection for a fairer comparison with the
CHAMP cross-track winds. This has required a renumbering of the figures as shown in the table below:

| Original Figure Number | New Figure Number |
|---|---|
| 1 CHAMP solar max 2001-2003 Longyearbyen | 1 CHAMP solar max 2001-2003 Longyearbyen |
| 2 CHAMP solar max 2001-2003 Kiruna | 2 CHAMP solar max 2001-2003 Kiruna |
| 3 CHAMP solar max 2005-2007 Longyearbyen | 3 geometry for projecting FPI winds onto CHAMP cross-track direction |
| 4 CHAMP solar max 2005-2007 Kiruna | 4 CHAMP solar max 2005-2007 Longyearbyen |
| 5 FPI solar max and min 2001-2003 Longyearbyen | 5 CHAMP solar max 2005-2007 Kiruna |
| 6 FPI solar max and min 2001-2003 Kiruna | 6 FPI solar max and min 2001-2003 Longyearbyen |
| 7 CHAMP vs HWM87 and HWM90 and FPIs | 7 FPI solar max and min 2001-2003 Kiruna |

| | |
|---|---|
| measurements made by UCL + Alaska (1980) | |
| 8 CHAMP vs FPI for 2- <= Kp < 4+ for Longyearbyen and Kiruna | 8 CHAMP vs HWM93 and FPIs measurements made by UCL + Alaska (1980) |
| 9 frequency distribution of CHAMP/FPI for solar max and both Longyearbyen and Kiruna | 9 CHAMP vs FPI for 2- <= Kp < 4+ for Longyearbyen and Kiruna |
| 10 CMAT2 model demonstration of effects of changing viscosity | 10 frequency distribution of CHAMP/FPI for solar max and both Longyearbyen and Kiruna |
| 11 height profiles of CMAT2 model zonal winds and comparison with the red line emission intensity profile. | 11 CMAT2 model demonstration of effects of changing viscosity |
| 12 frequency distribution of Kp | 12 height profiles of CMAT2 model zonal winds and comparison with the red line emission intensity profile. |
| 13 CMAT2 zonally averaged winds at Longyearbyen and Kiruna | 13 frequency distribution of Kp |
| 14 global maps of CMAT2 zonal winds comparing winds at 240 km with height integrated winds | 14 CMAT2 zonally averaged winds at Longyearbyen and Kiruna |
| 15 | 15 global maps of CMAT2 zonal winds comparing winds at 240 km with height integrated winds |

Responding to Ref 1 comment 2 has also meant a re-numbering of the remaining equations:

| Original Equation Number | New Equation Number |
|---|---|
| 1 | deleted |
| 2 | deleted |
| 3 | deleted |
| 4 | 1 |
| 5 | 2 |
| 6 | **5** (corrected original number which was out of order) |
| 7 | 3 |

| 8 | 4 |
|---|---|

Some additional points we noted:

We noted that Figure 12 required some more explanation of the simulations of the height profile of the
zonal winds (lines 633-645):

"Figure 12 illustrates how ground-based FPIs make measurements of the neutral winds at 240 km altitude. The
left plot shows a height profile of the CMAT2 zonal mean zonal winds at the latitude of Longyearbyen. There
are 6 simulations to demonstrate the effect on the height profile of the zonal mean zonal winds when changing
the viscosity. CMAT2 uses a viscosity term that is the weighted mean divided by the scale height of two
coefficients of viscosity: the molecular viscosity $\mu_m$; and the turbulent viscosity $\mu_t$. The simulations represent a
comparison with the original molecular viscosity (dark blue). The other lines are for low (yellow - divided by
100) and high molecular viscosities (pink - doubled). The low and high turbulent viscosities are represented by
the Prandtl numbers 0.7 (red) and 100 (green), where 2 is the default value used in CMAT2; which is relevant
for the height at which gravity waves deposit momentum (Liu et al., 2013). The light blue line labelled "Mata"
is an intermediate profile. As can be seen, the molecular viscosity dominates in the thermosphere above 100 km
and at the altitudes where the FPI is measuring. The dark blue and yellow lines are representative of a vertical
slice of Figure 11 left and right, respectively, for the latitude of Longyearbyen."

Replaced Nov-Jan with DOY 300-65 in abstract and line 424. This is the correct range of DOY used in
the FPI selection criteria, as well as the CHAMP data.
Added the following:
i)       an extra affiliation for MF
ii)       data availability
iii)       Co-author Rosie Hood's recently awarded PhD thesis as a reference
Moved and consolidated description of red line emission profile and winds with respect to height to
follow Figure 12.

**Anonymous Referee #1:** Interactive comment on Ann. Geophys. Discuss., https://doi.org/10.5194/angeo-2019-57, 2019. ()

Dear Editor, I have reviewed the paper by Aruliah et al. titled "Comparing high-latitude
thermospheric winds from FPI and CHAMP accelerometer measurements." The paper
describes polar observations from FPIs and the CHAMP satellite. Observations from
different instruments are compared and discrepancies discussed. It is important to the
community to understand the differences between the FPI and CHAMP observed thermospheric
winds. CHAMP winds are known to be larger than the FPI measurements.
While the cause for such discrepancy may be unknown at the moment, at least, we
should know how the two sets data are different. Hence, the paper is very important
and should be considered for publication. It can be a good reference for future users of
the two data sets. The paper in the current form, however, has some significant issues need to be addressed.

**1. The paper somehow lost focus. It got into too many sub-topics: FPI (old) -FPI (new)
differences, HWM87-HWM90 differences. I think for the purpose of understanding the
CHAMP and FPI difference, we should avoid using very old data. If the topic were the
long-term trend, then we should examine long data string. The value of this paper is
on the CHAMP and FPI comparison.**

We beg to differ on this comment. It is important to provide a context and full argument for this important finding, i.e. that there is a serious discrepancy between the two methods of measuring thermospheric winds; but that the UCL FPI results are consistent with the U.Alaska FPI. Then the FPI procedure is discussed in section

6.2. However, we have added a sentence (lines 480-481): "These are interannual and inter-solar cycle discussions for a later paper."

The HWM87 and HWM90 plots have been replaced with a single plot using the later version HWM93.

The calibration of FPIs is an important section attempting to understand the discrepancy in the CHAMP-FPI
winds. The University of Alaska FPI measurements at Longyearbyen, made in the early 1980s, are used to show
that their FPI measurements are consistent in phase with the UCL FPI measurements 20 years later. They are
also of a magnitude closer to the UCL FPI than the CHAMP winds. Even then Hedin et al. (1991) noted that the
U.Alaska ground-based FPI zonal wind magnitudes in 1980 showed smaller magnitudes than satellites.

**2. The section L67-89 has some major issues. The thermospheric dynamics is governed
by the momentum, energy, and continuity equations. It cannot be expressed in
the formulas listed in the section. CHAMP's larger winds do not necessarily lead to
larger temperatures. Smaller FPI winds are not always connected to the lower temperatures.
We cannot use temperature to verify wind values.**

Agreed that this is a very simplified argument, but the purpose is to highlight the repercussions of
overestimating the neutral wind when dividing energy between heating and acceleration of the neutral gas.
But considering the referee's concerns, we have decided to remove lines 67-92 and replace with a general
comment on the partitioning of energy (lines 65-71):

"With incorrect scaling, there arises a problem of distortion of energy budget calculations of the upper atmosphere. A precise estimation of energy supply to the system is hindered essentially, because the partitioning of kinetic and thermal energy channels becomes obscured. The acceleration of the neutral air in 3-D space with respect to the active driver of the plasma motion is important to estimate, for instance, the Joule heating rate as one of the most important thermal energy inputs. This has a knock-on effect on the calculation of the absolute density of the gas, which is an important parameter used in, for example, satellite orbit calculations."

Removal of these paragraphs and 3 equations has required re-numbering of the remaining equations as
outlined in the table above.

**3. I am lost in the section L279 – L316. I understand that the CHAMP cross-track**
**winds are not aligned with the eastward direction on the ground. The angle is different**
**for the ascending and descending nodes. The angles are +/- 7.2 deg and +/-13.3 deg**
**for KEOPS and Longyearbyen (ascending and descending). So the obvious thing to do**
**is to compare separately the ascending and descending nodes. Use the ground based**
**meridional and zonal components to compute the wind value along the direction of**
**CHAMP cross-track measurement (mostly to add the contribution from the meridional**
**winds). I don't understand why that was not done. I do not see an argument to let**
**me believe that I can ignore the viewing angle difference between the ground based**
**zonal wind and CHAMP cross track winds. So I suggest the authors use the ground**
**based FPI data to form the wind values along the CHAMP cross-track direction to do**
**a direct comparison. Or alternatively, using the CHAMP ascending and descending**
**node measurements to remove the contribution from the meridional wind and compute**
**the zonal wind. Hopefully, that would give you better comparison between the FPI and**
**CHAMP observations.**

The FPI vector winds have been projected on the cross-track direction for the ascending (East - alpha) and
descending (East + alpha) as suggested. The discrepancy in magnitude is still large. An extra Figure (new Fig 3)
has been added to illustrate the geometry.

**4. Figure 7 should be the focus of the paper. I really don't see much value having two HWM model runs**

**results shown here. You only need one. The FPI (Alaska) probably does not add much and more likes a**

**distraction. The old and new FPI comparison should be discussed in a different paper, you can have inter-**

**annual variations here.**

The two HWM models have been replaced with the later HWM93 model. The FPI (Alaska) data remains because it shows that the UCL FPI data are consistent with the Alaska FPI under fairly similar f10.7 and mean Kp. Indeed Hedin et al (1991) had noticed a systematic discrepancy between satellite and ground- based FPI winds even then.

**5. The solar minimum data are not very useful since there are no CHAMP observations.**

The original Figs 3 and 4 (now called Figs 4 and 5) show CHAMP data from solar minimum.

**6. I think the conclusion should be clearer that the CHAMP winds are overestimated.**
**As the paper points out that the CHAMP winds are almost the same magnitude as ion**
**drift at Kiruna, which is incorrect.**

Agreed.  This point is discussed in Section 6 for various aspects such as the role of viscous action
in the upper thermosphere (Sect. 6.1) or the ion-neutral comparisons (Sect. 6.4). To accentuate
this set of problems, we emphasized this with a expanded paragraph referring to the Fiori et al
(2016) Swarm measurements of ion velocities (lines 767-774), and two additional sentences at
the end of the Conclusions (lines 834-837). This is a key point for further discussions and also
for more modelling/simulation studies (lines 837-839).

"Fiori et al (2016) compared ion velocities measured by the Electric Field Instrument on Swarm with the CS10

statistical ionospheric convection model by Cousins and Shepherd (2010) which is based on 8 years of data (1998-2005) collected by 16 SuperDARN coherent scatter radars. The climatology represented by the CS10

model in Fiori et al's Figure 3a indicates speeds in the few hundreds of ms$^{-1}$, while the instantaneous values along the Swarm satellite pass (their Figure 3d) show much stronger drift peak values on the resolution level of seconds or shorter. Even after allowing for offsets, their 1-sec resolution corrected cross-track ion drifts achieve horizontal velocities well over 1,000 ms$^{-1}$, which probably indicates the highly dynamic behaviour in the auroral regions compared with quasi-stable conditions used for empirical models. However, recently Koustov et al.

(2019) compared the Swarm cross-track ion drifts with the SuperDARN radar network and found that the

Swarm ion velocities are a factor of 1.5 larger. They suggest reasons for the disparity, including refining the calibration of Swarm and the differences in spatial/temporal resolution."

"We may also need to rethink the procedure of comparing different spatial and temporal resolutions of in-situ satellite versus remote ground-based FPI measurements in terms of the geometry of cross-track winds at high latitudes."

**7. While Aruhiah et al. 2005 reference is listed for the instrument information. It will be**
**a great help to give a short paragraph on the two FPIs (gaps, aperture size, detector,**
**from which year to which years) given that instrument upgraded over the years.**

Added to section 5.2.

**Minor**

**1. In the comparison section, the FPI (Alaska) was used, but there is no mention of it**
**in the abstract and earlier instrument description. It should be added to the abstract, if**
**it is to be used. Personally, I think the data should be dropped.**

The U.Alaska FPI measurements at Longyearbyen illustrate that the UCL FPI measurements 20 years later are
consistent. They are a reference measurement, so do not need to appear in the abstract.

**2. Many the figures have very low resolution and are difficult to read.**

Figures 5-8 have been increased in size on the page. In particular, Figs 5 and 6 have been split so that a,b and
c,d are on 2 separate pages. (These are now re-labelled Figs 6-9)

**3. Figure 5, the FPI from one direction has error bars and the other does not. Why?**

Only one set of error bars were included for the sake of clarity of the plots. However, we have now included
the error bars for all the FPI look directions.

**4. L 444 to LL448. 'During the 1980s and 1990s we used state-of-the-art UCL designed**
**and built Imaging Photon Detectors (McWhirter et al., 1981) and then EMCCDs (Andor**
**iXon 887/885) were installed around 2005 (McWhirter, 2008). The revolution over the**
**last 30 years in : : :' Was the UCL FPI running with the imaging photon detector from**
**2001 to 2003, it is not clear in the paper. That is why I ask for a more detailed instrument**
**description to be added.**

We have added some more explanation in section 5.2, in particular the dates of changes of etalon. The etalon gap is important in the calibration of the measured Doppler shift to wind calculation, which is described in section 6.2. The effect of the detector is to improve the sensitivity to the photon counting by electon multiplying. This reduces the error of measurement. It does not change the calibration of the wind speeds (lines 445-459).

"During the 1980s and 1990s we used state-of-the-art UCL designed and built Imaging Photon Detectors (McWhirter et al., 1982). Astrocam Antares cameras replaced the IPD in the Svalbard FPI from 1998, and in the KEOPS 630 nm FPI in 2002. However, these cameras had the disadvantage of slow readout times which were essential for the best noise performance and so time resolution was compromised. In 2003 the first Electron Multiplying CCDs revolutionised low light level imaging. These cameras combined superior signal to noise ratio with very fast readout times. The first one was put into service at KEOPS in 2003, followed by Svalbard in 2005 (McWhirter, 2008). The huge advancement over the last 30 years in low light detectors has allowed atmospheric gravity wave observations using exposure times as little as 10 seconds at auroral latitudes (Ford et al., 2007). Note that the upgrade of the detector is to improve the photon sensitivity which reduces the error of measurement. It does not change the calibration of the wind speeds.

Any changes of etalon required re-calibration of the measured Doppler shift to calculate winds, as discussed in section 6.2. The KEOPS FPI used a 10 mm etalon gap up to January 2002, when it was replaced with an 18.5 mm gap etalon. Then in January 2003 a 14 mm etalon was put in, which has been there until the present time. For the Longyearbyen FPI there was a 14 mm etalon until April 2005, which was replaced with an 18.5 mm etalon from September 2005 until the present time."

**5. I do not see Harris 2001 paper (L554) in the reference. Or the reference date is wrong? It is not 2017, should it be 2001?**

Yes, my typo mistake. The reference is to the Harris PhD thesis in 2001.

**6. Equ. 5 is wrong. It does not match the two references.**

The coefficient of viscosity is based on Dalgarno and Smith (1962), where it is given as viscosity = $3.34 \times T^{0.71}$ micropoise

Equation 5 is the conversion to SI units. The Banks and Kockarts reference is removed.

**7. There should be some discussion on hypothesis C (L520).**

Hypothesis C – the assumptions of the FPI and CHAMP measurement techniques are discussed in sections 6.1-6.3. A sentence has been added to make this explicit.

**8. Why is the HWM model run not included in Figure 8 comparison for Kiruna?**

The HWM93 model has been run for both Longyearbyen and Kiruna and appears in the renumbered Figs

8 and 9.

We have removed old Fig 8a (Longyearbyen 15 min averages) and instead the renumbered Fig 9 shows only the Kiruna zonal winds as 1 hour averages, to match the Longyearbyen 1 hour averages shown in the renumbered Fig 8.

**Anonymous Referee #2** Interactive comment on Ann. Geophys. Discuss., https://doi.org/10.5194/angeo-2019-57, 2019. ()

This paper statistically compares upper thermospheric F-region winds measured by
two high-latitude ground-based Fabry-Perot Interferometers (one located near Kiruna
and other at Longyearbyen) and derived from in-situ accelerometer measurements onboard
the CHAMP satellite. One of the ground-based stations is located in the auroral
zone whereas the other one is in the polar cap. Results show that CHAMP winds are
systematically 1.5-2 times larger than FPI winds. Further, the authors utilize the existing
modeling tools for exploring the various possible reasons responsible for these
systematic discrepancies in winds obtained from in-situ and optical techniques. Overall,
this study can serve as an important reference for data users of these instruments.

In my view, the manuscript is loosely written. There is some repetitiveness of some of
the text and the manuscript could be streamlined quite a bit. I would strongly recommend
the authors to make clear, elaborate, and explain the following parts:

**1. Please explain the purpose of having fist four figures (Figures 1-4). I think they
are irrelevant and can be dropped without impacting the focus of the paper. Instead, it
would help focusing this study on the core topic - FPI and CHAMP wind comparison.**

The CHAMP figures 1-4 and FPI Figs 5-6 will be valuable to modellers to show phase and amplitude of the
seasonal variation of thermospheric winds in the polar cap and auroral region for the European sector. Note
these have been renumbered as in the table at the top. The FPI can only measure night-time winter winds. This
is the reason why satellite measurements are so important.

**2. Line 17: should be kinematic viscosity instead of viscosity?**

Molecular viscosity dominates in the thermosphere, so this has been made explicit throughout.

**3. Line 25: +-2 degrees in latitude, longitude, or both? Please explain.**

The radial distance is the horizontal equivalent of +/- 2 deg in latitude (i.e. ~220km horizontal radius) at 240
km altitude.

**4. Line 148: In Table 1 (column 4 and row 2), you mean 1.860 UT?**

Yes, thank you. Corrected

**5. Line 172: Emmert 2006a reference is not valid here because it is a climatological
study.**

Ok, thanks, point taken. Emmert ref removed

**6. Lines 299-317: The simplest and most direct way to compare CHAMP and ground
station winds would be to project ground station winds along the CHAMP cross track
winds; it is doable because both the zonal and meridional winds exist for ground station
FPIs.**

This has been done for Figures 7 and 8 (renumbered 8 and 9) and to determine the ratios of CHAMP/FPI along
the cross-track direction shown in the histogram in Fig 9 (renumbered 10).

**7. Figure 5 and 6: Please keep the figure titles consistent. Subfigures a/b titles are
not consistent with c/d titles: one shows Kp index in title and others not. In addition,**

**please keep consistency when using plus or minus symbols in Kp values. For example,**
**sometimes the manuscript uses Kp<2 and the other times Kp<2- [[or Kp<2o (line 367,**
**413, etc.) which may be a typo]]. Kp<2o is also present in Figures 6a and b. Moreover,**
**I would suggest using an actual math symbol (ï˝C ˘c) instead of <=.**

The <= has been replaced and the titles made consistent.

**8. Lines 424-426 are referred to which figure/figures?**

This is clarified in the text: ("The general trends seen in the northern winter CHAMP zonal winds (Figures 1-4, renumbered 1,2,4,5) are also seen in the FPI winds (Figures 5-6, renumbered 6-7). The phases match extremely well for both sites, however, there is a considerable difference in magnitude. The next 2 figures 7-8 (renumbered

8-9) show direct comparisons of CHAMP and FPI winds along the cross-track direction.")

**9. Figure 7:**
**- This comparison is done for Kp 2-4, whereas earlier figures and discussion was focused**
**on Kp 0-2. Same is true for Figure 8. Please explain the reason for this gear shift.**

There is a lot of modelling effort into studying the active ionosphere-thermosphere, so we wanted to show this
too. In particular it is relevant to the comparison with high latitude ion velocities discussed in section 6.4.

**- Please explain why HWM87 and HWM90 were used instead of HWM14? HWM14 is**
**the latest version of this empirical wind model.**

HWM93 is used to replace HWM87 and HWM90

**10. Figure 9: In addition to this figure, a plot showing CHAMP/FPI ratio as a function**
**of UT or LT would be really helpful.**

This is a very good idea, and has been added to Fig 9 (renumbered 10)

**11. Lines 518-522: The major source of discrepancies could be the assumptions used**
**when applying different wind extraction schemes as they can fail under different conditions.**

Indeed, this is the point of sections 6.1-6.3. Sentence added to make this explicit just before section 6.1 (lines
554-555):
"With respect to hypothesis C – the assumptions of the FPI and CHAMP measurement techniques are discussed in the following sections 6.1-6.3."

**12. Line 556: Please verify the viscosity expression.**

The coefficient of viscosity is taken from Dalgarno and Smith (1962). Here it is given as viscosity = $3.34 \mathrm{x} T^{0.71}$ micropoise

Equation 5 is the conversion to SI units. The Banks and Kockarts reference is removed.

**13. Lines 715-722: Project FPI wind vector along the CHAMP cross track wind component.**

Done, including an additional illustration of the geometry (figure 3). Words added, including (lines 752-754): "

To deal with this the UCL FPI zonal winds observed to the East and West are projected onto the CHAMP

ascending and descending cross-track directions, and then averaged into 1-hour bins, thus replicating the

CHAMP zonal wind averages (see Figures 7 and 8)."

**14. Section 6.4: I did not get the motive of adding this section. So, please state**

**explicitly the contribution of this section in this investigation.**

This is a very important argument. At high latitudes the average neutral wind at high latitudes is expected to be smaller than the ion velocities, since the latter are driven by the magnetospheric dynamo. This is stated in the 2nd sentence of this section.

---

## Author Response (AR2)

[revised manuscript text omitted]

2014.

**Response to reviewers 2nd set of comments**

We thank the two reviewers for taking extra time for a second look at our paper and make the following comments and changes from their suggestions for minor corrections.

Response to Reviewer 1

**The revised paper has improved greatly. However, I still have some issues in the current version. Since I did not raise those issues earlier, I am not going to insist revisions to the paper. With the exception of the Equ 2, I think should be revised.**

**1) I still don't understand the rational for including the U.Alaska FPI data in the discussion here.**

**"The calibration of FPIs is an important section attempting to understand the discrepancy in the CHAMP-FPI winds. The University of Alaska FPI measurements at Longyearbyen, made in the early 1980s, are used to show that their FPI measurements are consistent in phase with the UCL FPI measurements 20 years later. They are also of a magnitude closer to the UCL FPI than the CHAMP winds. Even then Hedin et al. (1991) noted that the U.Alaska ground-based FPI zonal wind magnitudes in 1980 showed smaller magnitudes than satellites."**

**I don't think you need almost 40-year old U.Alaska data to prove the UCL FPI data are of good quality. You have many publications in the past using the UCL FPI data, which should be sufficient to prove you have good data set. Most of FPIs do not have 40-year data to compared with.**

Thank you for the supportive statements about the UCL FPI dataset, this is appreciated.

As we state in the last paragraph of the Introduction, and again in the first paragraph of the Discussion section, there are various possible reasons for the discrepancy between the FPI and CHAMP winds. Either a calibration problem with one or the other; or that horizontal thermospheric winds vary with height above 250 km altitude; or the different spatial and/or temporal resolutions are crucial and inherent for the differing (in-situ versus remote)

measurement techniques. The point of showing the U.Alaska FPI data is to confirm that there is no anomalous offset in the UCL dataset, which is augmented by explaining the procedure and assumptions used in determining wind speeds from line-of-sight measurements (section 6.2 FPI Doppler shift to wind speed procedure).

2) **Here you still have another loose end. "Even then Hedin et al. (1991) noted that the U.Alaska ground-based FPI zonal wind magnitudes in 1980 showed smaller magnitudes than satellites. "**
**Thirty years ago, we did not have accelerometer data to compare with. Hedin et al. were talking about the comparisons between the wind data from DE-2, which is a combination of the FPI and WATS in-situ observation. If there were discrepancy between ground-based and satellite observations then, that was a totally different comparison with different technologies. It is hard to relate that discrepancy with the one at hand.**

Again we are trying to check whether it is a method/calibration problem, or a real height dependence. We have added the following sentence to emphasise this, and hopefully spark further investigations.

"However, it is also possible that the fact that two different methods of satellite measurements gave winds systematically larger than ground-based FPI measurements might support the existence of a vertical structure in the upper thermosphere."

3) **The paper is a little bit unbalanced. The paper is more focused on possible issues related to the FPI but not so much with CHAMP data. It should be obvious that the issue is with CHAMP. Perhaps, the authors should at least promise something to be done in future studies.**

Section 6.3 (CHAMP cross track wind procedure) already aims to indicate the efforts that have gone into refining the derivation of winds from drag. This section helps to highlight to the aeronomy community that the drag scaling factor is very difficult to determine. We have added the following sentence to the end of section 6.3:

"This difficulty is well known in the satellite engineering community, but perhaps less so in the aeronomy community."

and the final paragraph of the conclusion (section 7):

"But for satellite drag measurements, we note that satellite aerodynamic coefficients are difficult to determine absolutely"

**4) The Song (2009) paper does not add much to the discussion about the discrepancy. Same for the Vadas and Crowley's paper. Those two papers were not well explained. I leave the authors to decide to keep them or not.**

These two papers are relevant to provide other measurements indicating vertical structure, and suggestions of other possible mechanisms. So we have left them in, but the order of the paragraphs in which they appear has been reversed in order to improve the flow.

To end this section on the molecular viscosity of the upper thermosphere, two papers are referenced that report observations and suggest other possible mechanisms to support winds varying with height. Recently Vadas and Crowley (2017) published results from observations of 10 Travelling Ionospheric Disturbances at ~283 km altitude, observed in 2007 with the TIDDBIT ionospheric sounder near Wallops Island, USA. They used ray tracing on the TIDs and simultaneously measured a peak in the neutral wind at ~325 km altitude using a sounding rocket. They found a serious discrepancy between where the gravity waves were predicted to dump energy using conventional dissipative theory, and the observations from TIDDBIT and the rocket. Conventional theory predicted that all the gravity waves should have dispersed at a scale height below the rocket measurement. Consequently they have challenged convention and proposed that the molecular viscosity should not increase as rapidly with altitude above 220 km. This may account for some of the difference between the CHAMP and FPI zonal winds, but will need to be tested in future modelling studies.

The second paper by Song et al. (2009) states that the fastest acceleration of the neutrals occurs near 350 km in the F layer, where the effective neutral-ion collision frequency maximizes (see their Figs. 6 and 7). Considering the dynamic character of frequent changes of the IMF and the magnetospheric convection, the stronger accelerations at F2 layer heights could result in temporary vertical neutral wind gradients. However, the 1-D model approach neglects forces due to neutral pressure and effective molecular viscosity in the 3-D continuum of the upper thermosphere. To describe correctly the long-range coupling on time scales from longer than few seconds to less than 30 min, the inductive effect (Faraday's law) as well as the dynamic effect of the neutrals, (in particular acceleration terms), need to be considered (Song and Vasyliunas, 2013).

**5) The last sentence is pretty vague "We may also need to rethink the procedure of comparing different spatial and temporal resolutions of in-situ satellite versus remote ground-based FPI measurements in terms of the geometry of cross-track winds at high latitudes. "If you have issue with the CHAMP winds, then changing geometry is**

**not going to fix the problem.**

We have deleted the phrase "in terms of the geometry of cross-track winds" from this sentence and from the last sentence in the manuscript on lines 846-847. Comparing different resolutions of satellite and ground-based instruments concerns both measurement techniques.

**6) I still don't see how Equ 2 is the same as the authors showed in the reply. If you just**

**copy what you have in the reply, it will be fine. The formula in the paper is not clear.**

Added the following as suggested:

"which is the SI version, based on Dalgarno and Smith (1962) where it is given in units of micropoise."

**Overall, the revised manuscript is well written and logically organized. On the whole this is an important contribution on the optical vs in situ wind measurement discrepancy that is still to be resolved. I would strongly recommend this study for publication.**

**I have a couple of minor suggestions/comments.**

**1. Why data curves in the Figures 1, 2, and 4 have changed slightly compared to the original version? Are same data, as used in the original version of this paper, used here?**

The CHAMP data are now plotted in UT rather than LT. As a consequence the time intervals are shifted and comprise of different entireties with slightly different averages. The general behaviour is kept, of course.

**2. Why all the days and day numbers are changed in the paragraph between lines 476 and 483 in the revised manuscript?**

In the original manuscript the average monthly F10.7 data were calculated from Nov-Jan, but in the revised version the data were correctly recalculated to include Feb for each winter to correspond with DOY 300-65.

**3. Figure 7: 0<Kp<2o or 0<Kp<2-? See next comment**

**4. Line 370: 0<Kp<2o or 0<Kp<2- in the revised version?**

Re: points 3 and 4, it is already explained in the text that:

"The solar minimum data range is $0 \leq Kp < 2-$, but it was necessary to increase the geomagnetic activity spread for the solar maximum data to improve the statistics, so Figures 7a, b show $0 \leq Kp < 2o$."

**4. Line 525 of the revised manuscript: "average zonal wind" or "average cross-track wind"?**

Thank you for spotting this – should be "average cross-track winds"